

# Evaluation of the WRF lake module (v1.0) and its improvements at a deep reservoir

Fushan Wang [1,2], Guangheng Ni[1], William. J. Riley[2], Jinyun Tang [2], Dejun Zhu[1], Ting Sun[3]

[1]Department of Hydraulic Engineering, Tsinghua University, Beijing, China
[2]Earth and Environmental Sciences Area, Lawrence Berkeley National Lab, Berkeley, CA, USA
[3]Department of Meteorology, University of Reading, Reading, United Kingdom

*Correspondence to*: Ting Sun (ting.sun@reading.ac.uk)

William. J. Riley (wjriley@lbl.gov)

**Abstract.** Large lakes and reservoirs play important roles in modulating regional hydrological cycles and climate; however, their representations in coupled models remain uncertain. The existing lake module in the Weather Research and Forecasting (WRF) system (hereafter WRF-Lake), although widely used, did not accurately predict temperature profiles in deep lakes mainly due to poor lake surface property parameterizations and underestimation of heat transfer between lake layers. We therefore revised WRF-Lake by improving its (1) numerical discretization scheme; (2) surface property parameterization; (3)

diffusivity parameterization for deep lakes; and (4) convection scheme, the outcome of which became WRF-rLake (i.e., revised lake model). We evaluated WRF-rLake by comparing simulated and measured water temperature at the Nuozhadu Reservoir, a deep reservoir in southwestern China. WRF-rLake performs better than its predecessor by reducing the root-mean-square-error (RMSE) against observed lake surface temperatures (LSTs) from 1.4 °C to 1.1 °C and consistently improving simulated vertical temperature profiles. We also evaluated the sensitivity of simulated water temperature and surface energy fluxes to

various modelled lake processes. We found (1) large changes in surface energy balance fluxes (up to 60 W m$^{-2}$) associated with the improved surface property parameterization and (2) that the simulated lake thermal structure depends strongly on the light extinction coefficient and vertical diffusivity. Although currently only evaluated at the Nuozhadu Reservoir, we expect that these model parameter and structural improvements could be universal and therefore recommend further testing at other deep lakes and reservoirs.

# 1 Introduction

Inland waters such as lakes and reservoirs differ from their surrounding land with altered albedo, larger thermal conductance and heat capacity, and lower surface roughness, and therefore different radiative and thermal properties. These lake properties can exert significant influences on local and regional climate and are important in understanding lake-atmosphere interactions (Hostetler et al., 1994; Bonan, 1995; Lofgren, 1997; Krinner, 2003; Long et al., 2007; Samuelsson et

al., 2010; Dutra et al., 2010; Subin et al., 2012; Bin Deng et al., 2013; Xiao et al., 2016).



Lakes interact with atmosphere through energy, mass (mostly water), and momentum exchanges (Lerman et al., 2013). Generally, due to their larger thermal inertia and smaller roughness (Samuelsson et al., 2010; Xiao et al., 2016), lakes tend to attenuate surface diurnal temperature variation and enhance surface wind compared to the surrounding land. The influences of lakes on regional climate vary by season (Subin et al., 2012). In the early winter and spring, lakes warm and moisten overlying

air masses, generating the so-called "lake effect" on precipitation manifested as heavier snowfall in downwind regions (Bates et al., 1993; Niziol et al., 1995; Scott and Huff, 1996; Zhao et al., 2012; Wright et al., 2013). In the early summer, reduced heat fluxes into the atmosphere are observed in some temperate and high latitude lakes because of their lower surface temperature and smaller roughness than the surrounding land (Lofgren, 1997; Krinner, 2003; Dutra et al., 2010). Also, anticyclones (cyclones) tend to be intensified in summer (winter) through their interactions with the Great Lakes (Notaro et

al., 2013; Xiao et al., 2016). During fall and early winter, when the lake surface is warmer than the overlying air, high-latitude lakes release the heat collected during summer to the atmosphere, reducing snow accumulation in the surface areas around the lakes (Long et al., 2007).

Since lakes strongly affect the lower boundary conditions of heat, water, and momentum dynamics in the upper atmosphere, predicting weather and climate in lake basins or lake-rich regions requires realistic lake representations in

numerical weather prediction models (NWPMs). In the last decade, many efforts have been made focusing on the development and analysis of lake modules in such models (MacKay et al., 2009; Bonan, 1995; Mironov et al., 2010; Dutra et al., 2010; Stepanenko et al., 2010; Subin et al., 2012; Subin et al., 2013; Stepanenko et al., 2013). Although there exist sophisticated lake models that account for detailed spatiotemporal dynamics of various lake processes, it is not a common practice to fully couple them with atmospheric models because of their high computational cost and prohibitive complexity for coupling (MacKay et

al., 2009).

In contrast, one-dimensional (1D) models have been widely used for coupling with atmospheric models, because they are sufficiently fast to facilitate long-term coupled simulations and have performed well in simulating seasonal and interannual variations of lake water temperature (Peeters et al., 2002). Depending on how they parameterize eddy diffusivity, these 1D lake models mainly fall into two categories (MacKay et al., 2009; Perroud et al., 2009; Martynov et al., 2010; Stepanenko et

al., 2010; Table 1): the Hostetler-type models (e.g., the Hostetler Model (Hostetler and Bartlein, 1990; Hostetler et al., 1994), Minlake (Fang and Stefan, 1996), SEEMOD (Zamboni et al., 1992), LIMNMOD (Karagounis et al., 1993), MASAS and CHEMSEE (Ulrich et al., 1995), CLM4-LISSS (Subin et al., 2012), and WRF-Lake (Gu et al., 2015)) and more sophisticated turbulence models (e.g., the bulk model of Kraus and Turner (1967), DYRESM (Imberger et al., 1978), PROBE (Svensson, 1978), GOTM (Burchard et al., 1999), SIMSTRAT (Goudsmit et al., 2002), and LAKE (Stepanenko and Lykosov, 2005;

Stepanenko et al., 2011)). The Hostetler-type models use parameterized eddy diffusivity to model vertical mixing in the lake body. The eddy diffusivity is dependent on surface wind speed and lake stratification and its parameterization follows that of Henderson-Sellers (1985), which is formulated under the assumptions of unstratified Ekman flow (Smith, 1979). Although the representativeness of this eddy diffusivity scheme for lakes has not been systematically evaluated, these models have been applied in numerous coupled simulations with RCMs (Bates et al., 1995; Hostetler and Giorgi, 1995; Small et al., 1999) and





GCMs (Bonan, 1995; Krinner, 2003). The more complex turbulence models use the $k-\varepsilon$ model of turbulence and parameterize eddy diffusivity based on the Kolmogorov–Prandtl relation. Thus, two additional equations are required for the turbulent kinetic energy ($k$) and its dissipation rate ($\varepsilon$). Although models from both categories have been intensively used in climate models, they share the potential and common drawback that the eddy diffusivity representations may be inappropriate when
temperature gradients are weak (MacKay et al., 2009).

In addition to the above two categories, bulk mixed layer models have also been developed, including FLake (a relatively simple 2-layer model based on similarity theory (Mironov et al., 2010)) as a typical example. However, it is hard for these oversimplified lake models to capture seasonal stratification and to simulate water temperature well in deep lakes. Thus, the performance of the bulk mixed layer models in climate-modelling studies is still limited.

WRF-Lake is a 1-D lake model that has enjoyed popular uses for modelling how lakes affect local weather and regional climate (Gu et al., 2015; Mallard et al., 2015; Gu et al., 2016; Xiao et al., 2016). It is an eddy-diffusion model adapted from the Community Land Model (CLM) version 4.5 (Oleson et al. 2013) with further modifications by Gu et al. (2015). However, some previous studies and our evaluation here suggest that WRF-Lake is insufficient in simulating deep lakes and reservoirs (Mallard et al., 2015; Xiao et al., 2016). To better represent thermodynamic processes of deep lakes and reservoirs
in regional climate simulations, we thus revised the WRF-Lake model. Our revisions fall into four categories: (1) including an improved spatial discretization scheme; (2) improving the surface property parameterization; (3) improving the diffusivity parameterization for deep lakes; and (4) improving the convection scheme to avoid unphysical mixing. We evaluated the improved lake model, WRF-rLake (i.e., a revised lake model), at the Nuozhadu Reservoir, a deep reservoir (up to 200 m deep) in southwestern China and conducted sensitivity experiments to evaluate the effect of dominant processes and parameters on
simulated lake water temperature and surface heat fluxes.





**Table 1.** Common 1D lake models.

| Category | Model Name | Main Features | References |
|---|---|---|---|
| Hostetler-type Model | Hostetler Model | based on eddy diffusion concepts to represent vertical mixing of heat | Hostetler and Bartlein (1990) |
| | Minlake | based on Hostetler model; treatments of topography, inflow and outflow, and biochemical processes | Riley and Stefan (1988) Fang and Stefan (1996) |
| | SEEMOD | based on Hostetler model; treatments of topography, inflow and outflow, and biochemical processes | Zamboni et al. (1992) |
| | LIMNMOD | based on MIT-Wind-Mixing Model and SEEMOD; suited for water quality forecasting | Karagounis et al., (1993) |
| | MASAS and CHEMSEE | based on Hostetler model; especially suited for investigating anthropogenic organic compounds in lakes | Ulrich et al. (1995) |
| | CLM4-LISSS | based on Hostetler model; straight-sided; treatments of snow and ice physics | Subin et al., (2012) |
| | WRF-Lake | based on CLM4-LISSS with some differences in lake surface properties and heat diffusivity | Gu et al., (2015) |
| More Complex Turbulence Model | the bulk model | based on $k$–$\varepsilon$ model of turbulence | Kraus and Turner, (1967) |
| | DYRESM | based on turbulence closure scheme; applies Lagrange coordinates; especially designed for reservoirs with a treatment of inflow and outflow | Imberger et al., (1978) |
| | PROBE | based on $k$–$\varepsilon$ model of turbulence; generally designed for boundary layers in the environment but applicable to lakes; treatments of topography, inflow and outflow; equipped with a 2D option | Svensson, (1978) |
| | GOTM | based on $k$–$\varepsilon$ model of turbulence; originally designed for oceans but applicable to lakes; can be easily coupled to a collection of biogeochemical models | Burchard et al., (1999) |
| | SIMSTRAT | based on $k$–$\varepsilon$ model of turbulence; applies an seiche excitation and damping model to compensate for mixing under thermocline | Goudsmit et al., (2002) |
| | LAKE | based on $k$–$\varepsilon$ model of turbulence; treatments of heat and moisture transfer in soil and snow cover; suited for lake-soil systems | Stepanenko and Lykosov, (2005) |
| Bulk Mixed Layer Model | FLake | based on similarity theory; a simple 2-layer model | Mironov et al., (2010) |





## 2 Model description and improvements

### 2.1 Description of the current WRF-Lake model

The WRF-Lake model is a mass and energy balance scheme which vertically divides the lake into 20-25 layers and solves the 1D heat diffusion equation. The lake includes 0-5 snow layers; 10 lake liquid water and ice layers (hereafter lake body); and 10 sediment layers (Subin et al., 2012). The lake body water content is assumed to be constant and sediment layers are fully saturated. An infinitely small interface is assumed between the first lake layer and the overlying atmosphere to calculate lake surface fluxes of heat, water mass, momentum, and radiation. After subtracting latent and sensible heat fluxes from the surface net all-wave radiation, the residual energy fluxes are set as the top boundary condition to solve the heat diffusion equation. Mixing processes in the lake body include molecular diffusion, wind-driven eddy diffusion, and convective mixing (Hostetler and Bartlein, 1990). We describe below the basic model structure to indicate model processes and parameters that were analyzed and modified in this study.

### 2.1.1 Surface energy balance

In the WRF-Lake model, the energy balance at the lake surface is:

$$S + L - G = H + \lambda E, \tag{1}$$

where $S$ is absorbed shortwave radiation, $L$ is net longwave radiation, and $G$ is downward heat flux into the lake, $H$ is sensible heat, and $\lambda E$ is latent heat. The units for all variables in equation (1) are W m$^{-2}$. In this equation, $S$ is regulated by albedo ($a$); $G$, $H$, and $\lambda E$ are largely dependent on the thermal conductivity of the top lake layer ($\kappa$), aerodynamic resistance for heat ($r_{ah}$) and that for vapor ($r_{aw}$), respectively.

### 2.1.2 Radiation transfer and absorption in the lake body

For unfrozen lakes, a fraction ($\beta$) of the incoming solar radiation is absorbed by the surface water within the first 0.6 m. The remaining solar radiation is absorbed by the lake body following Beer's law:

$$\phi = (1-a)(1-\beta)S_i e^{-[\eta(z-z_a)]}, \tag{2}$$

where $\phi$ (W m$^{-2}$) is the distribution of solar radiation with depth $z$ (m; positive downward), $\beta$ is set to a constant 0.4 (Oleson et al., 2010), $z_a$ is the base of the surface absorption layer (0.6 m), and $\eta$ is the light extinction coefficient (m$^{-1}$) which is calculated as a function of the lake depth according to Hakanson (1995):

$$\eta = 1.1925 d^{-0.424}, \tag{3}$$

where $d$ (m) is the lake depth.





### 2.1.3 The heat diffusion solution

After the surface energy balance and radiation absorption are calculated, the following 1D thermal diffusion equation is solved:

$$c_w \frac{\partial T}{\partial t} = \frac{\partial}{\partial z}\left(k \frac{\partial T}{\partial z}\right) - \frac{d\phi}{dz}, \tag{4}$$

where $c_w$ is the volumetric heat capacity of water (J K$^{-1}$ m$^{-3}$), $T$ is water temperature (K), $t$ is time (s), $z$ is depth (m), $k$ is the thermal conductivity (m$^2$ s$^{-1}$), and $\phi$ is the solar radiation as in equation (3).

The thermal conductivity in WRF-Lake comprises molecular diffusivity ($k_m$) and wind-driven eddy diffusivity ($k_e$). $k_m$ is a function of thermal conductivity and volumetric heat capacity of water ($1.433 \times 10^{-7}$ m$^2$ s$^{-1}$). $k_e$ follows the Henderson-Sellers parameterization (shown in Appendix) and is determined by wind speed at 2 m above the water surface, latitude dependent Ekman profile, and lake stratification dependent Brunt-Väisälä frequency (Henderson-Sellers et al., 1983; Henderson-Sellers, 1985). However, for deep lakes, e.g., Lake Michigan (Martynov et al., 2010), Subin et al. (2012) argued that increasing the eddy diffusivity by a factor of 10, and in some cases by a factor of 100, substantially improved simulated lake water temperatures and surface fluxes. Thus, in WRF-Lake, the eddy diffusivity for lakes deeper than 15 m was multiplied by a factor ranging from $10^2$ to $10^5$ depending on lake depth and surface temperature (Gu et al., 2015).

### 2.1.4 Convective mixing

The convective mixing module is executed after the heat diffusion equation solution and is identical to that of Hostetler's lake model, which is based on the assumption that no temperature instability can be sustained in the lake body. In particular, at the end of any numerical time step, if there is denser water overlying lighter water, convective mixing happens rapidly and the lake body from surface to the unstable layers is mixed (Hostetler and Bartlein, 1990).

### 2.2 Improvements by WRF-rLake

As described below, we modified the existing WRF-Lake model (and named the revision as WRF-rLake) by improving the spatial discretization scheme, parameterization for surface properties and vertical diffusivity, and the convection scheme.





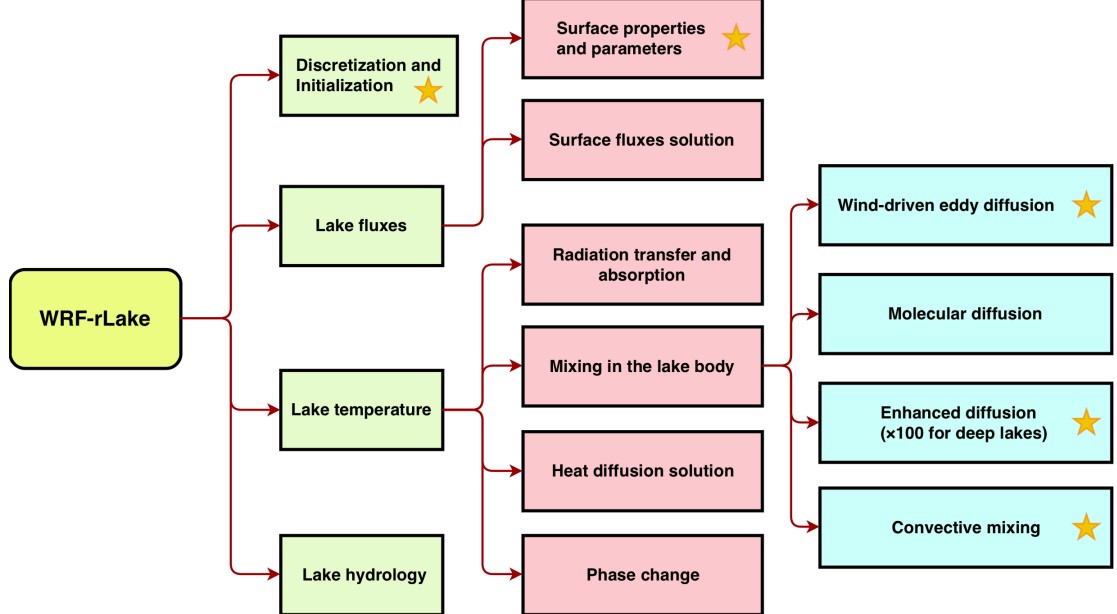

**Figure 1.** Flow chart of the WRF-rLake model. The yellow star indicates the process is modified or newly added.

### 2.2.1 Vertical discretization

The WRF-Lake model discretizes the water body into 10 layers with the top-most layer fixed to 0.1 m (Gu et al.,
2015) and each of the other nine layers constituting 10% of the total depth. Although this discretization works well with
shallow lakes (e.g., depth < 50 m), it may be problematic for deep lakes in two aspects. The first is depth loss, because the 9
layers below the first layer only account for 90% of the lake body, which, for very deep lakes, may cause up to 10% lake depth
loss and will potentially lead to energy loss of the simulated lake system. The second is the large layer thickness transition
between the first two layers. In the case of the Nuozhadu Reservoir (~200 m deep), the default discretization results in a sharp
thickness increase from 0.1 m to 20 m (200 times) between the top two layers, which may be numerically problematic.

We therefore introduced an improved discretization scheme for the lake body. When a lake is less than 50 m deep, a
new 10-layer discretization scheme is applied where the top layer is fixed at 0.1 m and the remaining depth is allocated evenly
among the remaining nine layers. For lakes deeper than 50 m, a 25-layer discretization scheme is used where the topmost layer
is set to 0.1 m and the remaining layers have their thicknesses increasing exponentially by a fixed factor (FF) that depends on
lake depth (Table 2). For the Nuozhadu Reservoir, FF is taken to be 1.29, which results in a series of layer depths (m) from
the top to the bottom of: 0.1, 0.1, 0.2, 0.2, 0.3, 0.4, 0.5, 0.6, 0.8, 1.0, 1.3, 1.6, 2.1, 2.7, 3.5, 4.6, 5.9, 7.6, 9.8, 12.6, 16.3, 21.0,
27.1, 35.0, 44.7 m.



**Table 2.** Fixed factors (FF) for vertical discretization for different lakes based on depth.

| Depth (m) | 50~55 | ~65 | ~75 | ~90 | ~105 | ~120 | ~145 | ~170 | ~190 | ~235 |
|---|---|---|---|---|---|---|---|---|---|---|
| FF | 1.20 | 1.21 | 1.22 | 1.23 | 1.24 | 1.25 | 1.26 | 1.27 | 1.28 | 1.29 |
| Depth (m) | ~275 | ~320 | ~380 | ~440 | ~520 | ~600 | ~700 | ~800 | ~1000 | >1000 |
| FF | 1.30 | 1.31 | 1.32 | 1.33 | 1.34 | 1.35 | 1.36 | 1.37 | 1.38 | 1.39 |

### 2.2.2 Surface properties and parameters

As discussed in section 2.1.1, the aerodynamic resistances for heat ($r_{ah}$) and vapor ($r_{aw}$) heat fluxes are functions of momentum ($z_{0m}$) and scalar roughness lengths ($z_{0h}$ for sensible heat and $z_{0q}$ for latent heat) and are critical for surface energy balance predictions. In WRF-Lake, $z_{0m}$ is set to 1 mm, 5 mm and 2.5 mm for unfrozen lakes, frozen lakes without snow, and frozen lakes with snow, respectively; while $z_{0h}$ and $z_{0q}$ are always kept equal to $z_{0m}$.

However, for unfrozen lakes, the momentum roughness length of 1 mm is often too large. Open seas are believed to have larger roughness lengths than lakes due to better developed surface moving waves, while the Engineering Sciences Data Unit (ESDU) (1972) documentation suggested $z_{0m} = 1$ mm for normal and 0.1 mm for calm seas. A number of lake studies also have shown 1 mm to be a maximum $z_{0m}$ value. For example, measurements on Lake Washington show $z_{0m}$ is generally below 1 mm and ranged between 0.01~1 mm (Ataturk and Katsaros, 1999); measurements on Lake Ngoring, a high-altitude lake in the Tibetan Plateau, found $z_{0m}$ ranged between 0.001~1 mm and seldom went beyond 1 mm (Li et al, 2015). Thus, in order to produce more realistic roughness lengths for lakes, CLM4-LISSS parameterized $z_{0m}$ based on the forcing wind, friction velocity, fetch, and lake depth. Adopting these parameterizations has produced more accurate surface heat fluxes and LSTs over many natural lakes (Subin et al., 2012).

We therefore adopted the CLM4-LISSS parameterization of roughness lengths with some further modifications:

- $z_{0m}$ is set to 2.4 mm for frozen lakes with snow and 1 mm for frozen lakes without explicit snow layers; and $z_{0h}$ and $z_{0q}$ are computed as functions of $z_{0m}$ and friction velocity $u_*$ (m s$^{-1}$).

- For unfrozen lakes, $z_{0m}$ is parameterized as:

$$z_{0m} = max\left(\frac{\gamma \nu}{u_*}, \alpha \frac{u_*}{g}\right), \tag{5}$$

where $\gamma$ is a dimensionless empirical constant (0.1), $\alpha$ is the dimensionless Charnock coefficient described below, and $g$ is the acceleration of gravity (Fairall et al., 1996; Charnock, 1955; Smith, 1988):

$$\alpha = \alpha_{min} + (\alpha_{max} - \alpha_{min})\, exp[-min(A, B)], \tag{6}$$

$$A = \left(\frac{Fg}{u^2}\right)^{1/3} \Big/ f_c, \tag{7}$$

$$B = \varepsilon \frac{\sqrt{dg}}{u}, \tag{8}$$





where $F$ (m) is the lake fetch (assumed to be 25 times of lake depth when observations are unavailable), $u$ (m s⁻¹) is 2 m wind speed, $d$ (m) is lake depth, $\alpha_{min} = 0.01$, $\alpha_{max} = 0.11$, and $\varepsilon = 1$. $A$ and $B$ account for influences from fetch and depth, respectively.

We also made further improvements in WRF-rLake's roughness lengths parameterizations. In LISSS, $f_c$ should be 100 but was tentatively set to 22, corresponding to the use of $u$ instead of $u_*$ in equation (7). Subin et al. (2012) recommended that the future lake models should relax this assumption by setting $f_c = 100$ and directly applying $u_*$ rather than $u$, which we have done in WRF-rLake. Because $u_*$ depends on surface roughness lengths, which in turn depend on $u_*$, we introduced a fixed-point iteration for the equations relating $u_*$ and surface roughness lengths.

### 2.2.3 Improved diffusivity parameterization

The Henderson-Sellers parameterization (see Appendix) for wind-driven eddy diffusivity underestimates mixing in deep lakes and was therefore increased in WRF-Lake by a multiplicative factor (Gu et al., 2015). However, this treatment may trigger new problems. As $k_e$ declines exponentially with depth (equation A1), it is more likely to be underestimated in deeper layers than in the topmost one. Thus, enlarging $k_e$ by the same factor for the whole lake may introduce new problems in two ways.

First, $k_e$ may be overestimated in lake surface layers. A number of empirical studies have estimated the effective vertical heat diffusivity in lakes and coastal oceans using heat flux measurements and tracer distribution measurements. These studies showed that vertical heat diffusivity in natural lakes seldom exceeds ~1 cm² s⁻¹ and in oceans, where the forcing winds are usually stronger and moving surface waves are better developed, vertical heat diffusivity exceeds ~10² cm² s⁻¹ (Hutchinson, 1957; Li, 1973; Kullenberg et al., 1973; Kullenberg, 1974; Jassby and Powell, 1975; Sarmiento et al, 1976; Quay et al., 1980).

We thus set a maximum of 10² cm² s⁻¹ for wind-driven eddy diffusivity (corresponding to the maximum value for open seas) to avoid overestimation in surface layers.

Second, for deep layers, $k_e$ may still be underestimated by WRF-Lake because $k_e$ is forced to decline exponentially with depth. Additional diffusion terms should be included to account for other sources of turbulence in deeper layers where wind driven eddies cannot penetrate. In WRF-rLake, we adopted the enhanced diffusion term ($D_{ed}$) from CLM4-LISSS (Ellis

et al., 1991; Fang and Stefan, 1996; Subin et al., 2012), which was originally suggested by Fang and Stefan (1996) to compensate for unresolved turbulence, even below ice or at large depth:

$$D_{ed} = K(N^2)^{-0.43} , \qquad\qquad\qquad (9)$$

where $K$ is a lake dependent parameter ($1.04 \times 10^{-8}$) and $N$ (s⁻¹) is the Brunt-Väisälä frequency, with $N^2$ suggested to be limited by a minimum of $7.5 \times 10^{-5}$ s⁻² (Fang and Stefan, 1996). However, as discussed by Subin et al. (2012), this suggested value

for $D_{ed}$ is only several times larger than $k_m$ and may also be too small for deep lakes. We therefore evaluated an increase in $K$ by a factor of 100 for lakes deeper than 50 m and argue that more analyses are required to robustly represent unresolved turbulence.





### 2.2.4 Convective mixing

In WRF-Lake, density instability is the prerequisite for convective mixing. As we have mentioned above, density for each water layer is first calculated based on lake water temperature, then adjacent layers are compared for their densities to decide whether there should be convective mixing. However, since the water density is calculated as a function of temperature,

when the temperature gradient between two adjacent layers is small (usually less than $10^{-3}$ K m$^{-1}$, but still with a lighter layer overlying a heavier layer), small numerical errors may incorrectly trigger convective mixing. In WRF-rLake we therefore set a density gradient threshold of $10^{-4}$ kg m$^{-3}$ m$^{-1}$ (which is equivalent to a temperature gradient threshold of about $10^{-3}$ K m$^{-1}$) to avoid this unphysical convective mixing. In our tests at Nuozhadu Reservoir, convective mixing can penetrate as deep as 5 meters below the surface in WRF-Lake, while it is almost always restricted within the top 1 m in WRF-rLake.

**3 Data and Modelling Experiments**

### 3.1 Study area

The Nuozhadu Reservoir is near the downstream end of a group of reservoirs along the Lancang River (22˚38' N, 100˚26' E), which is located in southwestern China and is called the Mekong River when it leaves China and enters Laos. The Nuozhadu Reservoir has a dam height of 262 m and a normal water level of 812 m above sea level. The water depth upstream

of the dam in year 2015 is around 200 m. The water surface area has increased more than 10 times after construction of the reservoir. Owing to its particular location, great depth, and large surface area, the Nuozhadu Reservoir serves as a good example for research on the impacts of artificial inland waters on regional climate.





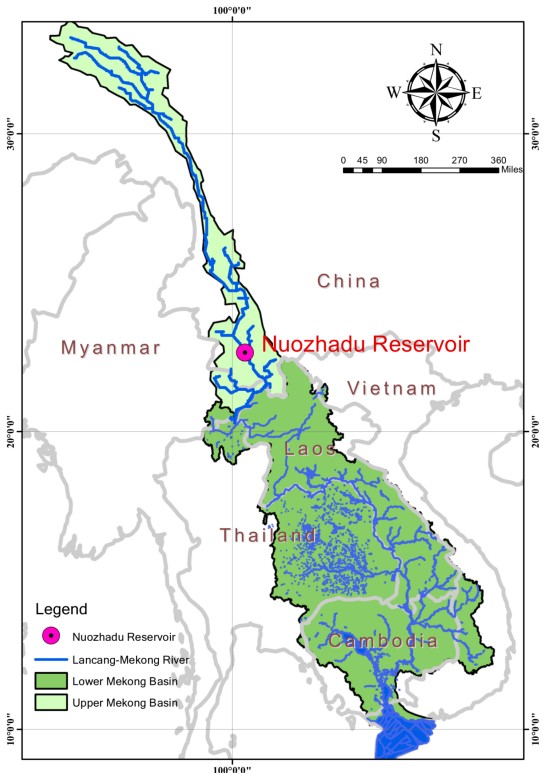

**Figure 2.** Location of the Nuozhadu Reservoir.

### 3.2 Forcing data

Our study period covers 1 January 2015 to 31 December 2015, a year when the reservoir was under normal operation.

5    The downward shortwave radiation and downward longwave radiation (W m$^{-2}$) were obtained from the China Meteorological

Forcing Dataset (Kun et al, 2010; Chen et al, 2011) with a temporal resolution of 3 hours and spatial resolution of 0.1° × 0.1°.

A linear interpolation was applied to these data to obtain hourly forcing for WRF-rLake.

Other forcing data, including atmospheric temperature (K), atmospheric pressure (Pa), atmospheric specific humidity

(kg kg$^{-1}$), atmospheric wind speed in the east and north directions (m s$^{-1}$), and precipitation (mm s$^{-1}$), were acquired with a

10    temporal resolution of 1 hour from a meteorological observatory run by China Huaneng Group Co., Ltd., the construction unit

of the Nuozhadu Reservoir.   The station (22°40′N, 100°23′E) is located about 5 km upstream (or northwest) of the Nuozhadu

Dam with an observational height of 10 m.





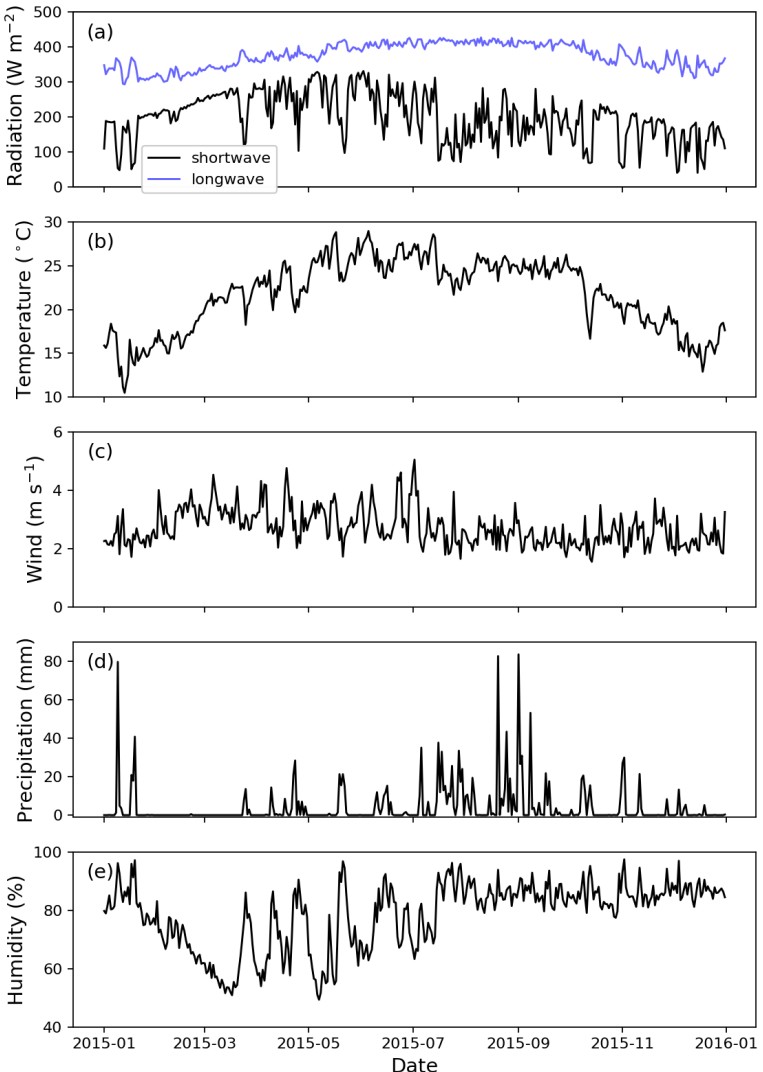

**Figure 3.** The year-long meteorological forcing data used include (a) shortwave and longwave radiation, (b) air temperature, (c) wind speed, (d) precipitation and (e) humidity. All data are averaged to produce mean daily values.

### 3.3 Observations for model initialization and evaluation

Water temperature was measured hourly from 712 m to 804 m above sea level at an interval of 2 m on the Nuozhadu dam during 2015 (Figure 4). Since the reservoir is in a tropical region, surface water temperature is higher than 20 ℃ throughout



the year. In 2015, the water level was dropped from January to June in preparation for the rainy season and rose gradually thereafter from July to December.

Measured water temperatures on 1 January 2015 was used to initialize the first 90 m of the water body. The remaining 110 m of depth (for which observations were not made) were initialized using simulated results at the same site on the same

5    day by Delft3D FLOW (Chen, 2017), a three-dimensional hydrodynamic model proven to be sufficiently accurate at the Nuozhadu Reservoir.

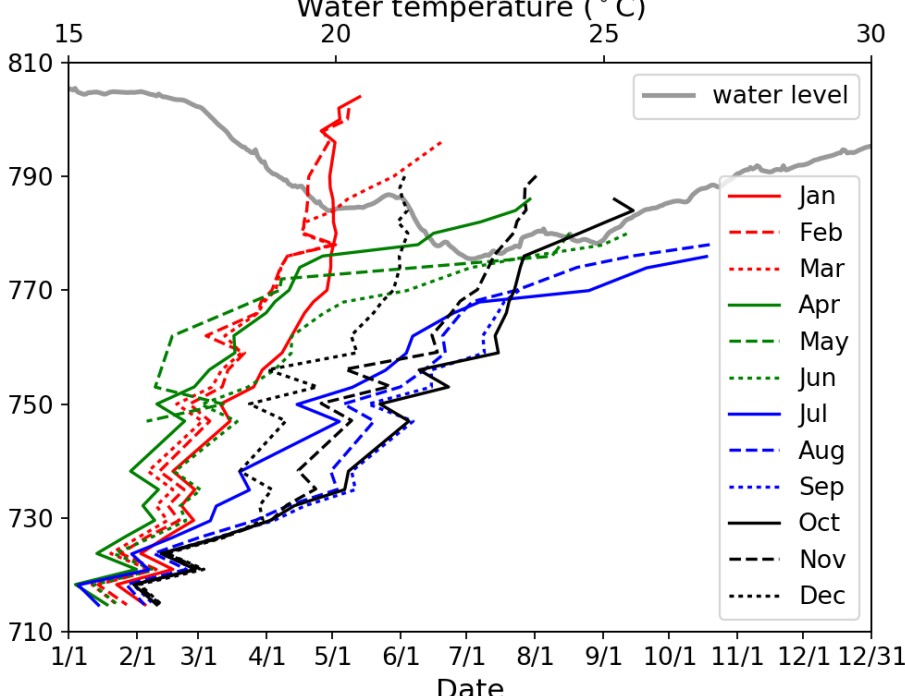

**Figure 4.** Measured monthly water temperature profile for Nuozhadu Reservoir in year 2015. The light gray line indicates the water level variation throughout the year.

**3.4 Numerical experiments**

To examine the incremental improvements in the WRF-rLake simulations, we performed four sets of numerical experiments analyzing four key parameterizations (Table 3):

1) Vertical discretization ("Lyr" set): the default WRF-Lake 10-layer and modified 25-layer settings were contrasted to assess the impacts of different vertical discretization schemes.

2) Vertical Diffusivity ("Diff" set): we examined the impact of vertical diffusivity by applying different diffusivity schemes: the original scheme by Hostetler and Bartlein (1990), the scheme by Gu et al. (2015), the scheme by Gu et al. (2015) with





enhanced diffusion term, and the scheme as discussed in section 2.2.3 or called "modified" diffusivity as adopted by our new model WRF-rLake.

3) Roughness length ("Rou" set): momentum and scalar roughness lengths are set to 1 mm, 10 mm, calculated as in Subin et al. (2012), or calculated with further modification as in section 2.2.2, to examine the effects of different roughness lengths schemes.

4) Light extinction coefficient ("Ext" set): through model tests, we conclude that in addition to the schemes we modified, the light extinction coefficient is also a key parameter for accurately modelling deep lakes (Hocking and Straskraba, 1999). Thus, we tested the impacts of light extinction coefficient for values of 0.13 $m^{-1}$ (default), 0.30 $m^{-1}$, 1.00 $m^{-1}$, and 3.00 $m^{-1}$. We concluded that the best performance could be achieved by increasing the light extinction coefficient to ~1.00 $m^{-1}$, which thus is adopted by our baseline run (BL).

In addition, a control run (CTL) was configured with default roughness lengths, extinction coefficient, and vertical diffusivity (Gu et al. 2015) as in the default WRF-Lake; and a baseline run (BL, i.e., our proposed new model structure) was configured with all modifications described in section 2.2 and a tuned light extinction coefficient to demonstrate the effects of each improvement in WRF-rLake.

**Table 3.** An overview of numerical experiments designed to demonstrate sensitivity in WRF-rLake.

| Experiments | Roughness Lengths | Extinction Coefficient | Diffusivity | Vertical Discretization |
|---|---|---|---|---|
| *Reference runs* | | | | |
| CTL | 1 mm | 0.13 $m^{-1}$ | Gu et al. (2015) | 25 layers |
| BL | Subin et al. (2012) and modified | 1.00 $m^{-1}$ | modified | 25 layers |
| *Sensitivity runs* | | | | |
| Lyr_1 | Subin et al. (2012) and modified | 1.00 $m^{-1}$ | modified | 10 layers |
| Lyr_2* | Subin et al. (2012) and modified | 1.00 $m^{-1}$ | modified | 25 layers |
| Diff_1 | Subin et al. (2012) and modified | 1.00 $m^{-1}$ | Hostetler and Bartlein (1990) | 25 layers |
| Diff_2 | Subin et al. (2012) and modified | 1.00 $m^{-1}$ | Gu et al. (2015) | 25 layers |
| Diff_3 | Subin et al. (2012) and modified | 1.00 $m^{-1}$ | Gu et al. (2015) +enhanced term | 25 layers |
| Rou_1 | 1 mm | 1.00 $m^{-1}$ | modified | 25 layers |
| Rou_2 | 10 mm | 1.00 $m^{-1}$ | modified | 25 layers |
| Rou_3 | Subin et al. (2012) | 1.00 $m^{-1}$ | modified | 25 layers |
| Ext_1 | Subin et al. (2012) and modified | 0.13 $m^{-1}$ | modified | 25 layers |
| Ext_2 | Subin et al. (2012) and modified | 0.30 $m^{-1}$ | modified | 25 layers |
| Ext_3 | Subin et al. (2012) and modified | 3.00 $m^{-1}$ | modified | 25 layers |

*: configuration of *Lyr_2* is the same as *BL*.



## 4 Results and discussion

### 4.1 Comparison of simulated temperature fields between WRF-Lake and WRF-rLake

In comparing WRF-Lake simulations (CTL) to the observations (Figure 5), the LSTs are underestimated most of the time with a Mean Bias Error (MBE) of -0.69 °C. The largest bias is found in the 1st quarter where it reached about -2 °C. This

bias mainly resulted from the overestimation of roughness lengths by prescribing them to 1 mm, which resulted in too large outgoing latent and sensible heat fluxes (section 4.2).

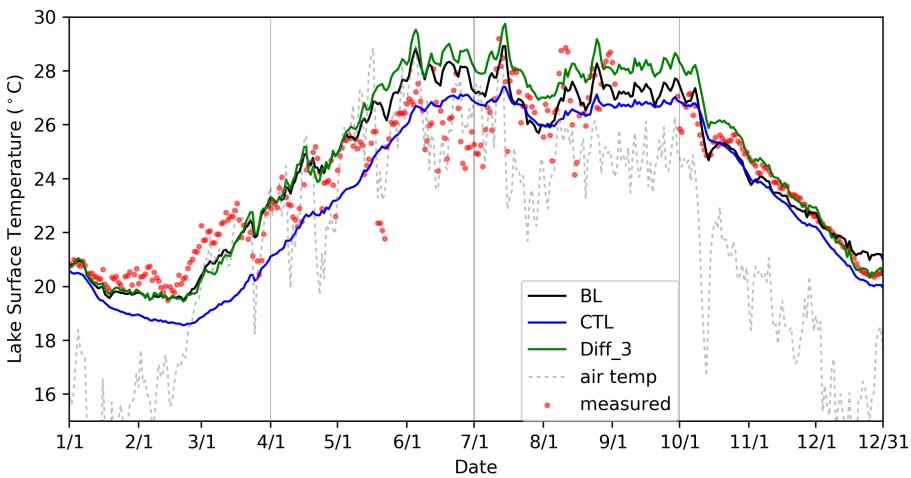

**Figure 5.** Time series of lake surface temperatures by observation (red dots), baseline (BL: black line), control (CTL: blue line), and Diff_3 (green line), and air temperature (grey dashed line) of Nuozhadu Reservoir for the period 1 January 2015 through 31 December 2015.

The configuration Diff_3 (with "half" modified diffusivity) overestimated LSTs by ~1.3 °C, reaching an MBE of 0.61 °C. Seasonally, the LSTs are very well reproduced in the 1st and 4th quarter but are overestimated in the 2nd and 3rd quarter of the year. The vertical temperature profile (Figure 6) shows that in the 2nd and 3rd quarters, Diff_3 simulated too much vertical mixing in the top 10 meters, resulting in warmer water temperatures in this zone. Meanwhile, a sharp temperature decline is observed between 10 and 20 m depth followed by an underestimation of temperature below 20 m, suggesting too weak vertical

mixing simulated below 20 m. Further discussion in terms of diffusivity is shown in section 4.3.





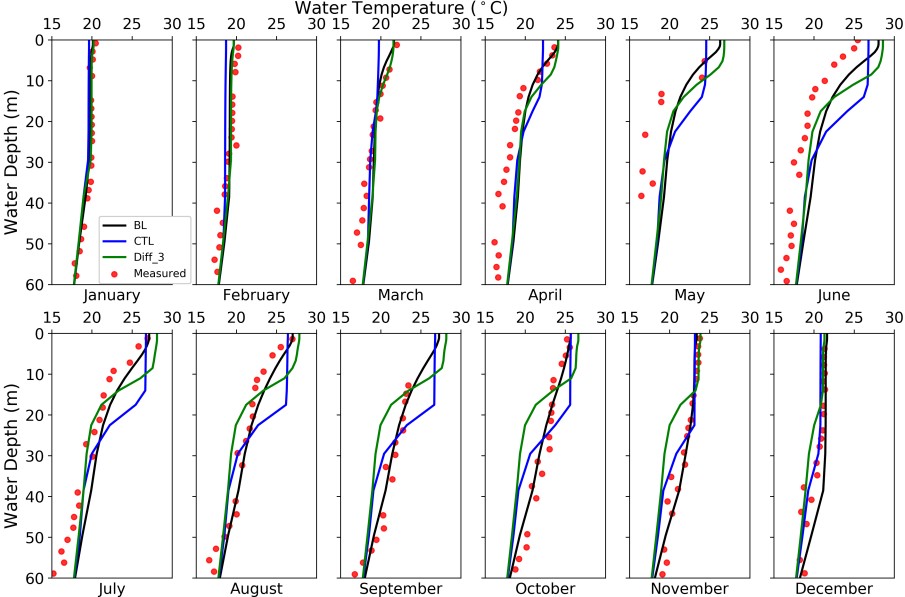

**Figure 6.** Monthly vertical temperature profile for the first 60 m water in year 2015 by observation (red dots), BL (black line), CTL (blue line) and Diff_3 (green line).

The simulation by BL is better in terms of RMSE of simulated LSTs, which is reduced to 1.14 °C from 1.35 °C by

5    CTL. Vertical temperature profiles were also improved, as the thermocline in the top 10 meters is reproduced in hot seasons, in contrast to the CTL and Diff_3 simulations. More detailed statistical metrics of the vertical temperature profile (Table 4) suggest that BL gives the best simulations among the three simulations compared.

**Table 4.** Statistics of the discrepancy between simulated (CTL, Diff_3, and BL) and observed LSTs and monthly temperature profiles during year 2015. Root Mean Square Error (RMSE) and Mean Absolute Error (MAE) are calculated between each simulation and measurement.
10    Mean Bias Error (MBE), Max Bias, and Min Bias are computed by simulation minus measurement. Coral and green shades indicate the largest and smallest values among three simulations, respectively.

|  |  | CTL | Diff_3 | BL |
|---|---|---|---|---|
|  | RMSE (°C) | 1.35 | 1.45 | 1.14 |
|  | MBE (°C) | -0.69 | 0.61 | 0.25 |
| LSTs | Max Bias (°C) | 3.31 | 5.55 | 4.57 |
|  | Min Bias (°C) | -3.37 | -2.00 | -1.86 |
|  | MAE (°C) | 1.09 | 1.00 | 0.83 |
|  |  |  |  |  |
|  | RMSE (°C) | 1.51 | 1.47 | 1.13 |
| Monthly | MBE (°C) | 0.48 | 0.32 | 0.57 |
| Temperature | Max Bias (°C) | 6.14 | 5.63 | 3.39 |
| Profile | Min Bias (°C) | -2.31 | -3.58 | -1.37 |
|  | MAE (°C) | 1.10 | 1.10 | 0.84 |





Applying all the modifications to vertical diffusivity (discussed in section 2.2.4), we obtained the best simulations by BL; however, we note that the original diffusivity parameterization of Henderson-Sellers (1985) may be inappropriate for deep lakes. We suggest more thorough evaluation and modification to this parameterization should be carried out in future research.

**4.2 Effects of vertical discretization**

In these experiments, Lyr_1 adopted the default WRF-Lake 10-layer scheme and was identical to BL except for discretization. Lyr_2 (identical to BL) uses the modified 25-layer scheme. Overall, compared to Lyr_1, Lyr_2 reduced the RMSE against monthly observed lake temperatures profiles from 1.64 °C to 1.13 °C. Lyr_1 predicted too much vertical mixing in the top 10 m, failing to reproduce the thermocline in this zone in summer (Figure 7). In the first two months, the difference in top 10 m between Lyr_2 and Lyr_1 was not obvious. But as the lake warms up, the thermocline in Lyr_2 (top 10 m) is

intensified, hindering heat transfer to the underlying layers, which in turn further strengthened the thermocline. However, this process in not captured by the Lyr_1 model, rather a well-mixed water column in the top 10 m (top two layers) is simulated throughout the year. From March to June, the overmixing by Lyr_1 also resulted in colder LSTs, which in turn produced lower sensible and latent heat fluxes by up to 10 W m$^{-2}$ and 20 W m$^{-2}$, respectively (not shown). Lower outgoing heat fluxes kept more energy stored in the lake body, explaining why the whole lake body became increasingly warmer throughout the summer

compared to Lyr_2. This underestimation of outgoing heat fluxes was reversed when the surface temperature in Lyr_1 finally exceeded that of Lyr_2 and produced more sensible and latent heat fluxes to the atmosphere from October to December.

      In summary, the 10-layer scheme produced more uniform temperature fields across the water column and, due to its coarser spatial discretization, poorly predicted the thermocline where temperature changes rapidly. We further tested a 100-layer scheme but its simulations did not differ significantly from the 25-layer scheme (not shown here), so we conclude that

the 25-layer scheme is sufficient for deep lakes like the Nuozhadu Reservoir.





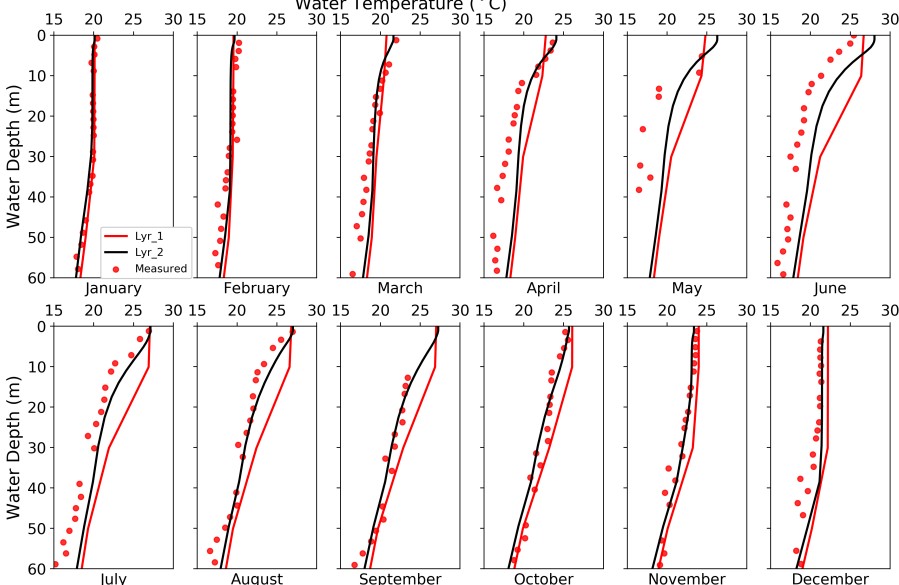

**Figure 7.** Monthly vertical temperature profile for the first 60 m water by observation (red dots), Lyr_1 (red line) and Lyr_2 (black line) in year 2015.

### 4.3 Effects of diffusivity

5        BL, Diff_1, Diff_2, and Diff_3 form a group of sensitivity experiments for diffusivity. In the case of the Nuozhadu Reservoir, Diff_2 applies the eddy diffusivity of Diff_1 with an increase of 100 times throughout all layers as is in Gu et al. (2015); Diff_3 additionally includes the enhanced diffusion term on top of Diff_2.

For the monthly-averaged vertical temperature profile (Figure 8), the BL scheme best captures the decrease of water temperature with depth and LSTs. Diff_1 yields strong stratification within 10 meters of the surface, indicating that the eddy

10   diffusivity by Hostetler and Bartlein (1990) is too small and prevents heat from transferring from the surface to depth. This suppression reduces heat stored in the lake body and weakens thermal inertia of the lake, leading to biased high LSTs in summer and biased low LSTs in winter.





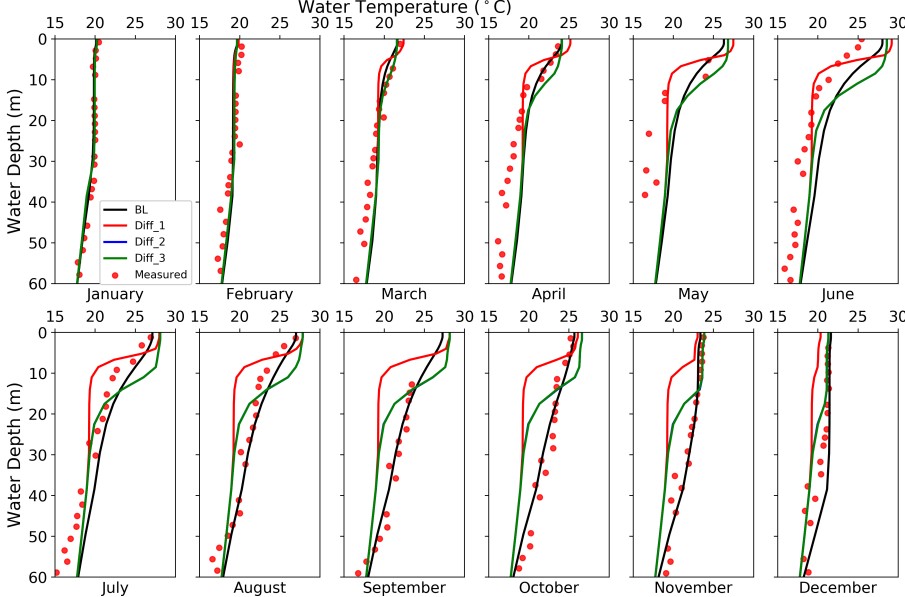

**Figure 8.** Monthly vertical temperature profile for the first 60 m water by observation (red dots), BL (black line), Diff_1 (red line), Diff_2 (blue line), and Diff_3 (green line) in year 2015. Note Diff_2 and Diff_3 overlaps each other.

       Diff_2 and Diff_3 produce identical vertical temperature profiles, indicating that the enhanced diffusion term is

5    relatively small compared to eddy diffusivity and makes little difference. Diff_2 and Diff_3 produce lake layers of almost

uniform temperatures in the top 10 m (Figure 8). This pattern is further confirmed by the strong vertical mixing in the first 10

m inferred by the vertical profile of overall diffusivity that includes molecular, eddy, and enhanced diffusivity (Figure 9). The

overall diffusivity by Diff_2 in the top 10 meters could be as large as $10^3$ cm$^2$ s$^{-1}$, which is too large compared to estimates in

the literature (section 2.2.3), supporting the necessity for limiting eddy diffusivity (section 2.2.4).





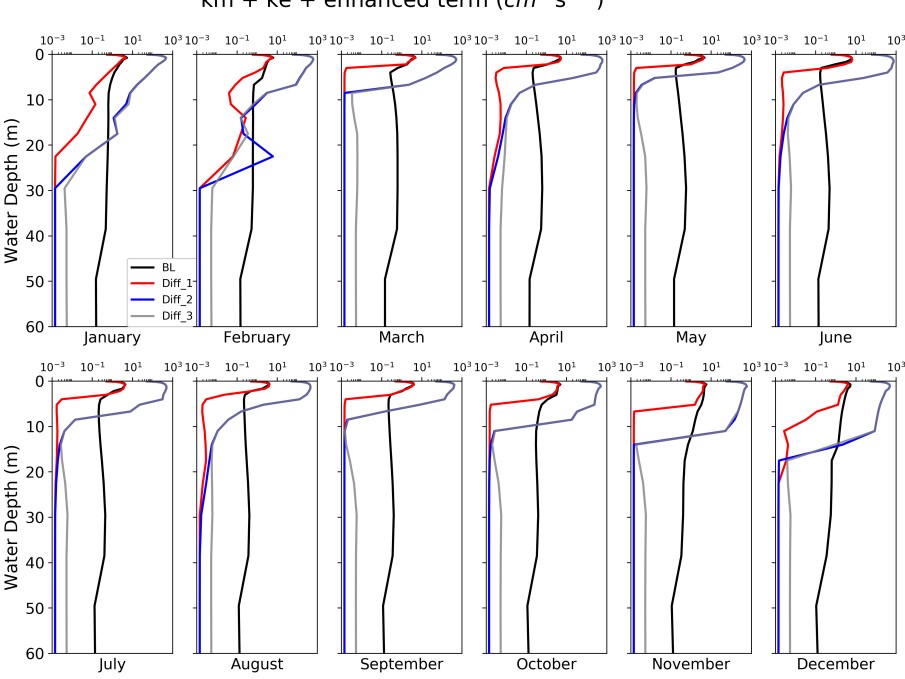

**Figure 9.** Monthly vertical diffusivity profile for the first 60 m water by BL (black line), Diff_1 (red line), Diff_2 (blue line), and Diff_3 (gray line) in year 2015.

The overall vertical diffusivity below 20 m by Diff_2 or Diff_3 is of the same magnitude as molecular diffusivity

($1.43 \times 10^{-3}$ cm$^2$ s$^{-1}$). By increasing the enhanced diffusion term by 100 times for deep lakes, BL yielded more reasonable vertical diffusivity below 20 m (Figure 9) which is very close to an estimation at a very similar lake: Li (1973) estimated the vertical diffusivity in Lake Zürich, a lake with similar topography and depth (~130 m deep) to the Nuozhadu Reservoir and concluded its value for vertical diffusivity ranged between $0.1 \sim 1$ cm$^2$ s$^{-1}$, with a mean diffusivity of $\sim 0.5$ cm$^2$ s$^{-1}$.

With the constrained eddy diffusivity and "enhanced diffusion term", BL produced the best temperature profiles

compared to observations (Figure 8). Further, the total diffusivity affects both the shape of vertical temperature profiles and the quantity of energy transferred down from the surface, which further influences LSTs.

### 4.4 Effects of roughness lengths

We next compare BL, Rou_1, Rou_2, and Rou_3, experiments that were conducted with different roughness length parameterizations, where the latter three have roughness lengths of 1 mm, 10 mm, and the parameterization from the Subin et

al. (2012) scheme. In BL, the value of roughness values varied but were almost always smaller than Rou_1 and Rou_2 (generally less than 0.5 mm). BL and Rou_3 produced almost identical LSTs (Figure 10) and latent and sensible heat fluxes





(Figure 11), suggesting that the modification added in section 2.2.2 (i.e., setting $f_c$ equal to 100 and applying $u_*$ rather than $u$ in equation 7), although being more realistic, do not influence the overall performance of WRF-rLake significantly. BL performs better in terms of RMSE of simulated LSTs, which is 1.14 °C compared to 1.34 °C by Rou_1 and 2.46 °C by Rou_2.

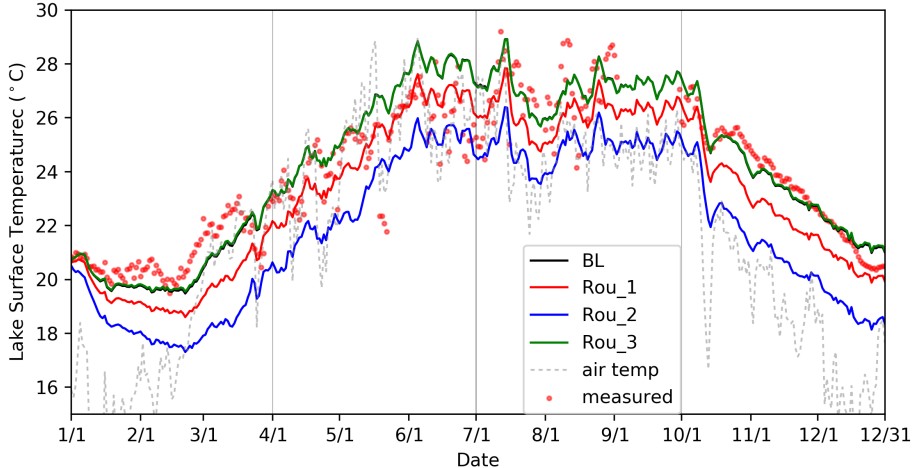

**Figure 10.** Time series of LST by observation (red dots), BL (black line), Rou_1 (red line), Rou_2 (blue line), Rou_3 (green line) and air temperature (grey dashed line) of Nuozhadu Reservoir in year 2015. Note that BL and Rou_3 overlap most of the time.

When the roughness lengths are fixed to 1 mm, an increase in sensible heat fluxes up to 30 W m$^{-2}$ is caused in cold seasons but slight decreases are resulted in warm seasons (Figure 11a). A larger effect is observed in latent heat flux, which is increased throughout the year by up to 30 W m$^{-2}$ (Figure 11b). Annual average LST is reduced by ~1 °C due to excessive outgoing heat fluxes, mainly the latent heat flux. These effects are amplified when fixing the roughness lengths at 10 mm, resulting in up to 100 W m$^{-2}$ increases in winter and 50 W m$^{-2}$ decreases in summer for sensible heat fluxes and up to 200 W m$^{-2}$ increases throughout the year for latent heat fluxes. With Rou_2, annual average LST is accordingly reduced by ~2.5 °C compared to BL.





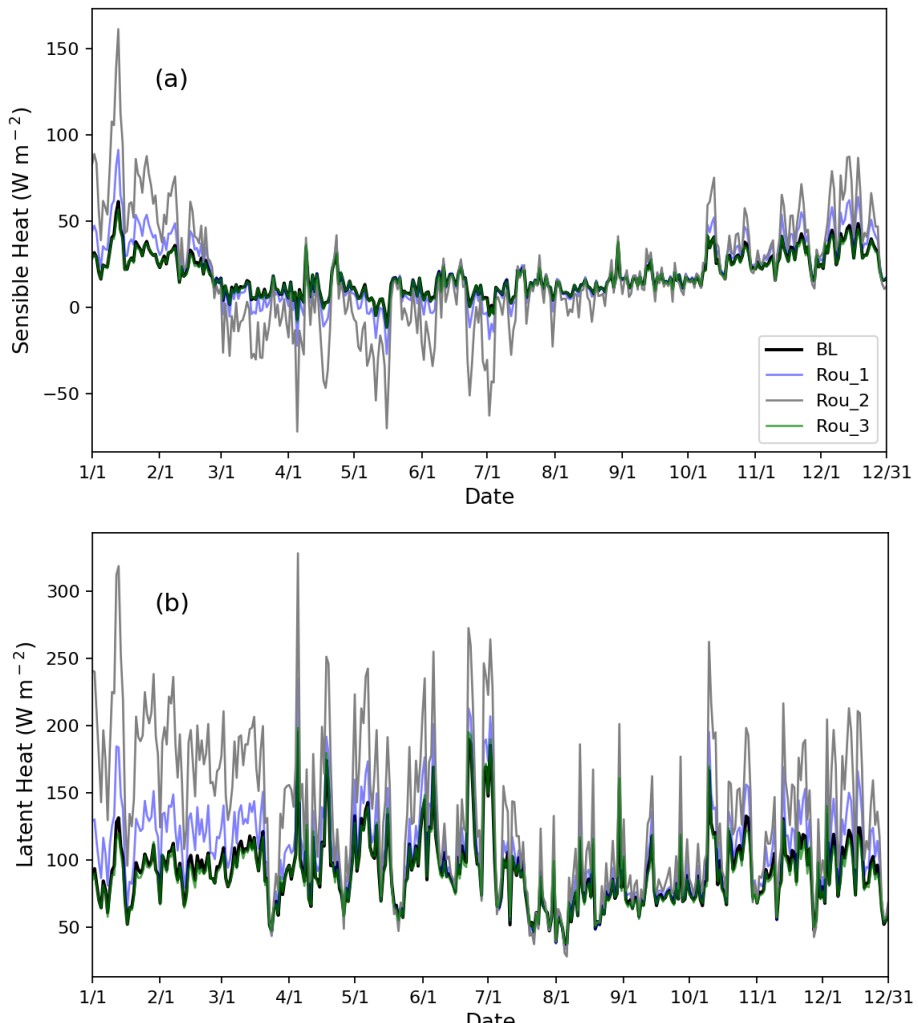

**Figure 11.** Time series of (a) upward sensible heat flux, (b) upward latent heat flux by BL (black line), Rou_1 (blue line), Rou_2 (gray line) and Rou_3 (green line) of Nuozhadu Reservoir in year 2015.

5   **4.5 Effects of light extinction coefficient**

Light extinction coefficient is the key parameter controlling the absorption and distribution of solar radiation in the lake body. BL, Ext_1, Ext_2, and Ext_3 form a group of experiments for extinction coefficient by prescribing it as 1.00 $m^{-1}$, 0.13 $m^{-1}$, 0.30 $m^{-1}$, and 3.00 $m^{-1}$ respectively. Although larger variability of observed values for light extinction coefficient





were reported (e.g., 0.05 to 7.1 m$^{-1}$ in Subin et al. (2012)), we chose values between 0.13 to 3.00 m$^{-1}$ in this study because minimal changes were predicted for values outside of that range.

The control of extinction coefficient over energy distribution is reflected in the monthly averaged vertical temperature profiles (Figure 12). In general, as the extinction coefficient increases, a larger portion of energy from solar radiation is maintained in the shallow layers, producing shallower stratification. As the extinction coefficient decreases, more solar radiation penetrates to depth, resulting in a better-developed epilimnion (the top-most and well mixed layer in a thermally stratified lake, occurring above the deeper hypolimnion). BL and Ext_3 produced very similar temperature profiles, which indicates that when the extinction coefficient is sufficiently large, increasing it merely brings in marginal influences in the vertical temperature profile. Specifically, for the Nuozhadu Reservoir, the threshold is ~1.0 m$^{-1}$.

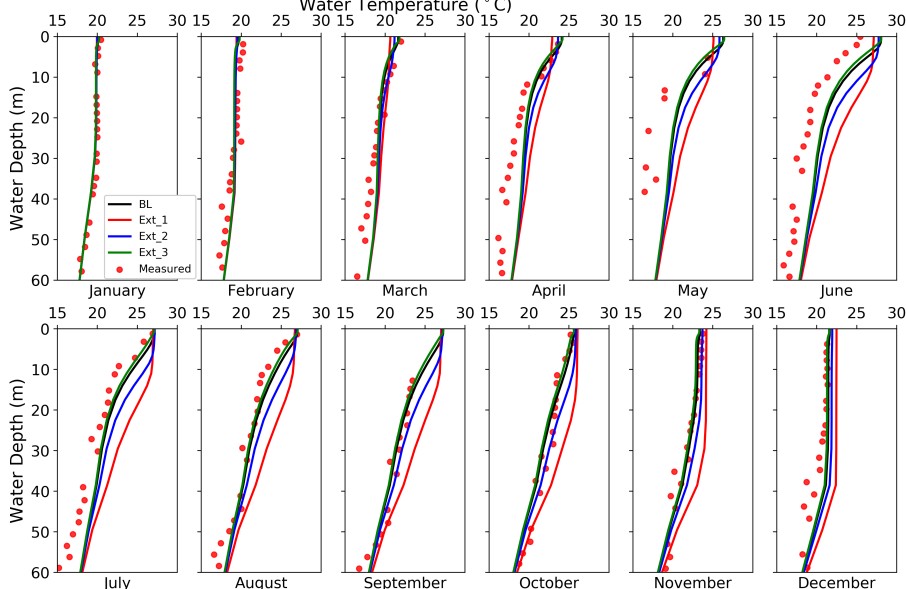

**Figure 12.** Monthly vertical temperature profile for the first 60 m water by observation (red dots), BL (black line), Ext_1 (red line), Ext_2 (blue line), and Ext_3 (green line) in year 2015.

In the four experiments, Ext_1 applied the default light extinction coefficient by WRF-Lake (the smallest). Solar radiation penetrated deeper and energy was distributed more evenly in the lake body with less temperature stratification in the topmost 10 m almost all year around. The accumulative influence of lower lake opacity is evident. In spring and summer, Ext_1 simulated lower LSTs than other scenarios as more solar radiation penetrated into and warmed up deeper layers. Further, with lower LSTs, less sensible and latent heat were dissipated and more energy was maintained within the lake body. By the end of the year, Ext_1 resulted in an integrally warmer water body than other scenarios with higher water temperatures of 1.5 – 2 °C.





We note that lake biology is a dominant factor influencing lake optical properties (Cristofor et al., 1994), which itself is affected by many processes, including lake hydrology and biogeochemistry. As modelling such processes is beyond the scope of the current WRF-Lake module, the parameterization of light extinction in WRF-Lake is simply based on lake depth (equation 3), which simplified the light absorption profile of the Nuozhadu Reservoir. Considering the substantial impact of
lake opacity on lake temperatures and surface fluxes, more sophisticated parameterizations of light extinction coefficient should be developed with considerations of lake hydrology, chemistry, and biology.

## 5 Conclusions

In this study, we revised the WRF-Lake model by adding a new spatial discretization scheme, modifying surface property and vertical diffusivity parameterizations, and adopting a revised convection scheme. The revised lake model, WRF-
rLake, was evaluated at the deep Nuozhadu Reservoir in southwestern China and demonstrated overall improved performance in simulating water temperatures in comparison with WRF-Lake, the current lake model in WRF. Compared to WRF-Lake, WRF-rLake reduced the RMSE against observed lake surface temperatures (LSTs) from 1.35 °C to 1.14 °C and that against monthly observed lake temperatures profiles from 1.51 °C to 1.13 °C.

Based on the comparison between WRF-Lake and WRF-rLake, we found that the coarse discretization of water layers
in the current WRF-lake made it less able to accurately predict temperatures in the thermocline. An adaptive 25-layer scheme was thus introduced to WRF-rLake and demonstrated better performance than the original 10-layer scheme by reducing the RMSE against observed lake temperatures profiles from 1.64°C to 1.13 °C.

In WRF-Lake, the lake surface roughness length is prescribed to be 1 mm, which we found could bias simulated surface fluxes and surface temperatures. Lakes have much smoother surfaces, and thus smaller roughness lengths, compared
to land and oceans, and lake roughness lengths vary with wind and surface waves. Replacing the original parameterization of roughness lengths with our proposed parameterization (equations 5,6,7,8) reduced latent heat flux by up to 30 W m$^{-2}$, and considerably improved LST simulations by reducing the RMSE against observations from 1.34 °C to 1.14 °C.

Simulations of temperature stratification and surface fluxes are sensitive to lake opacity, which modulates the absorption of solar radiation in the lake body. Considering that lake opacity may vary over more than two orders of magnitude
(Subin et al., 2012), more detailed global datasets on lake opacity based on remote sensing should be developed. Field measurements of extinction coefficients will be critical for achieving high quality weather and climate simulations in lake-rich areas.

Previous studies have recognized that the wind-driven eddy diffusivity parameterization by Hostetler and Bartlein (1990) is insufficient to simulate large and deep lake surface fluxes and water temperatures. Gu et al. (2015) therefore increased
eddy diffusivity for deep lakes with large multiplicative factors that depend on lake depth and surface temperature. However, we found such treatment resulted in unrealistically large eddy diffusivity in about the topmost 10 m of the Nuozhadu Reservoir and failed to compensate for the unresolved 3D mixing processes in deeper layers. WRF-rLake thus adopts a maximum of $10^2$





cm$^2$ s$^{-1}$ for eddy diffusivity to avoid overestimation in the surface layers and an enhanced diffusion term (further enlarged by 100 times for deep lakes), resulting in more realistic eddy diffusivities in deeper layers. In the case of the Nuozhadu Reservoir, adopting all the modifications to eddy diffusivity produced overall similar diffusivity to those measured in Lake Zürich (a lake with similar topography and depth). Although considerable improvements have been brought in by WRF-rLake, the

parameterization for vertical diffusivity can be further evaluated and improved to resolve remaining uncertainties in its performance at other lakes, especially for deep lakes.

Evaluation of the parameterizations, discretization, and sensitivity analyses recommended here for the deep Nuozhadu Reservoir should be performed at other lakes with different bathymetry, climate, etc. Overall, we found that simulations of lake temperatures and surface energy balance can be improved by WRF-rLake with respect to the discretization scheme, lake

opacity, parameterization for surface properties and vertical diffusivity, and the convection scheme. Our future work will focus on coupling between the WRF-rLake module and whole WRF system to further examine the online performance of coupled system.

*Code and data availability.* The WRF-rLake source code used in this study is archived at https://github.com/wfscheers/WRF-
rLake.git for discussion. Instructions for running the offline model can also be found in README.md via the above link. Data for forcing and initializing the model at the Nuozhadu Reservoir in year 2015 is available upon request to T. Sun (ting.sun@reading.ac.uk).

*Acknowledgments.* The study is supported by National Natural Science Foundation of China (NSFC) through Grants 51679119
and 91647107. This study is also supported by Huaneng Lancang River Hydropower Inc. J.Y. Tang and W.J. Riley are supported by the Director, Office of Science, Office of Biological and Environmental Research of the US Department of Energy under contract No. DE-AC02- 05CH11231 as part of the Regional and Global Climate Modeling (RGCM) Program. T. Sun is supported by NERC Independent Research Fellowship (NE/P018637/1).

**Appendix A: Parameterization for wind-driven eddy diffusivity**

In the Henderson-Sellers parameterization scheme, wind-driven eddy diffusivity ($k_e$) is determined by wind speed at 2 m above the water surface, latitude dependent Ekman profile, and lake stratification dependent Brunt-Väisälä frequency (Henderson-Sellers et al., 1983; Henderson-Sellers, 1985):

$$k_e = (ku_*z/P_0)e^{(-k^*z)}(1 + 37Ri^2)^{-1}, \tag{A1}$$

where $k$ ($= 0.4$) is von Karman's constant, $u_*$ is the surface friction velocity (m s$^{-2}$), $P_0$ ($= 1.0$) is the neutral value of the
turbulent Prandtl number, $k^*$ is a latitudinally dependent parameter of the Ekman profile, and $Ri$ is the gradient Richardson number.



The parameter $u_*$ is given by:

$$u_* = 1.2 \times 10^{-3}\, u \,, \qquad\qquad (A2)$$

where $u$ (m s-1) is wind speed at 2 m agl.

The parameter $k^*$ is expressed as:

$$k^* = 6.6 \times (sin\,\phi)^{1/2} u^{-1.84}\,, \qquad\qquad (A3)$$

where $\phi$ is the latitude of the lake being modeled.

The parameter $Ri$ is determined as:

$$Ri = \frac{-1 + \{1 + 40N^2 k^2 z^2 / [u_*^2 exp\,(-2k^* z)]\}^{1/2}}{20}, \qquad\qquad (A4)$$

where $N$ (s-1) is the Brunt-Väisälä frequency specified as:

$$N = [-g/\rho(\partial\rho/\partial z)]^{1/2}\,. \qquad\qquad (A4)$$

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
