# Peer review of "Evaluation of the WRF lake module (v1.0) and its improvements at a deep reservoir"

_Geoscientific Model Development, 2018_

## Referee Comment (RC1) · Anonymous Referee #1 · 18 Oct 2018

This manuscript presents a study striving to improve and calibrate an existing 1D lake model (parameterization) applied to a large artificial reservoir in China. Though the paper falls well into the journal scope, the main problems of the paper (lack of scientific novelty and methodological drawbacks) call for major revisiting of the whole study.

The base version of the model used is WRF-Lake, used in a number of studies during recent years (Gu et al., 2015; Gu et al., 2016; Xiao et al., 2016). The authors introduce into the model physical parameterizations already tested in other 1D models, especially CLM4-LISSS (Subin et al., 2012). Increasing the vertical resolution of the model is rather a technical improvement, as it is quite evident a-priori, that having 10 numerical layers is a rough resolution for deep lakes, where only 2-3 layers would cover the mixed layer. The authors follow the same approach as in (Gu et al., 2015; Gu et al., 2016;

[Figure]

Xiao et al., 2016) to tackle additional mixing in thermocline, which is just to multiply the molecular diffusivity by 100 or other large calibrated multiplier. This is the simplest way which does not take into account the evident physical effects like suppression of mixing by stratification. The approach by Fang and Stefan (1996) to parameterize background diffusivity takes into account stratification (eq. (9)), but the authors reject it. They argue, that original eq. (9) provides insufficient mixing. I would expect, that true development of the model physics would mean to replace primitive calibration of constant multiplier by calibration of constants in eq.(9) or likewise still simple but physically-sound parameterizations. Radiation parameterization (eq. (2)) assumes that shortwave radiation starts to decay with depth only below top 0.6 m, which is unphysical and easy to fix, assuming non-PAR (photosynthetically-active radiation) radiation to be absorbed at the surface, and PAR to be attenuated immediately below (and it is done in a such a way in almost all 1D lake models). I also see a notable drawback of the paper in that no empirical constrains have been involved on water turbidity for this particular lake. I can hardly imagine that no Secchi disk measurements have been performed at all. Treating both radiation extinction coefficient and background diffusivity which are the main controls for vertical distribution of heat as calibration parameters, you may attain similar vertical temperature distributions at different combinations of those parameters, and what would be the physical sense of that? The reservoir exhibits drastic surface level changes (about 30 m!), which would certainly influence the temperature profiles and introduce the vertical velocity in eq. (4), but the latter was not done, and the possible effects of level changes were not even discussed.

I suggest that this paper cannot be accepted and the authors should significantly change the methodology of their study prior to submitting the manuscript again.

References Gu, H., Jin, J., Wu, Y., Ek, M. B., & Subin, Z. M.: Calibration and validation of lake surface temperature simulations with the coupled WRF-lake model, Climatic Change, 129(3-4), 471-483, doi:10.1007/s10584-013-0978-y, 2015. Gu, H., Ma, Z. and Li, M.: Effect of a large and very shallow lake on local summer precipitation over the

Lake Taihu basin in China, Journal of Geophysical Research: Atmospheres, 121(15), 8832–8848, doi:10.1002/2015jd024098, 2016. Subin, Z. M., Riley, W. J., & Mironov, D.: An improved lake model for climate simulations: model structure, evaluation, and sensitivity analyses in CESM1, Journal of Advances in Modeling Earth Systems,4(1), -, 2012.

---

## Referee Comment (RC2) · Anonymous Referee #2 · 9 Nov 2018

Review for "Evaluation of the WRF lake module (v1.0) and its improvements at a deep reservoir", 2018, by Wang et al.

Overall comments:

The manuscript describes several modifications to the lake module within WRF, which are systematically included in a series of experiments, ultimately showing that these modifications result in improved model performance within a large, deep reservoir. Overall, these modifications are explained and justified well, and it is encouraging to see that they result in more accurate simulation of surface temperatures and in more realistic temperature profiles. I have identified mostly minor issues which are outlined below, and the paper should be accepted once these are properly addressed.

1. The one notable drawback to this paper is that there is no evaluation of simulated ice coverage, which can be a significant factor in how some lakes interact with the atmosphere. Although this follows naturally from the fact the that the reservoir being evaluated does not appear to experience freezing temperatures, this may limit the applicability of these results to large, deep lakes that do experience freezing, such as the Great Lakes. This limitation should be discussed.

2. It should be clarified early on that this evaluation of the lake model in WRF is done with observed forcing data, instead of model simulated fields. I understand that the authors' intent is most likely to evaluate the lake module free from bias that may be present in the WRF-simulated fields. However, as the model is referred to as WRF-Lake, readers may assume that the coupled system is being evaluated here. The need for such analysis isn't even mentioned until the last line of the paper, but would be better placed much earlier on.

3. Several figures (1, 2, 3) are never referenced in the text.

4. Here, observed water temperature profiles are used and the description of the experiments implies that no spin-up time was given to the model. This differs from the practice of many other modeling efforts where observed profiles of lake temperatures are not available, some of which use larger domains that include multiple lakes. In such applications, a sufficiently long spin-up would be the only way to obtain realistic temperature profiles. Clarify whether spin-up was used and discuss the implication for your results.

Specific comments:

1. P. 2, line 11: During this time of the year, snow is enhanced around the Great Lakes, not reduced.

2. P. 3: Works by Gula et al. (2012) and Mallard et al. (2014) (which coupled WRF to FLake in 1-way and 2-way model configurations, respectively) should be briefly mentioned alongside the discussion of the FLake model in the introduction, as it is the only other lake model that has been coupled with WRF, according to my knowledge.

3. P. 7, line 7, "approximately 10%" as 90% is included plus the 0.1 m first layer.

4. P. 8, first paragraph: Relationship between SH and LH and Zom is not well-explained in the earlier referred to section. Subin et al. (2012) contains equations that do relate the fluxes to aerodynamic resistance, and I suggest pointing readers to the appropriate section so they can find a more thorough discussion.

5. P. 8, line 18: This modification for frozen lakes does not appear well-justified.

6. P. 9, last paragraph: K is stated to be lake dependent, but a constant for it is then specified. Does K need to be provided in each lake or is it assumed to be equal to the provided constant? Also, clarify whether the Kx100 modification is applied everywhere in lakes deeper than 50 m or if it is only applied below 50 m.

7. P. 10. It is stated that this reservoir provides a good example of the impacts of artificial water bodies on regional climate, but this focus is not put in further context. Why did the authors choose to study an artificial body instead of a natural one?

8. P. 11 first paragraph: As the LW and SW data are interpolated from 3 hourly obs, peak radiation values may be underestimated. This should be stated in the text.

9. Fig. 4. Label the y-axis. Also, clarify that the "water level" shown (according to the inset box) is not actually the water level (which, having a mean of 812 m, does not seem to be consistent with the values shown here).

10. Table 3 "Roughness Lengths" column: I believe the constants given here refer to the roughness lengths for unfrozen lakes, based on previous discussion, but this should be clarified.

11. Figure 5 and other similar figs: The observed temperatures shown here were taken near the dam of the reservoir. Are the simulated LSTs taken and averaged

over a similar area or are they representative of lake-average conditions? If it's the later, then direct comparison to observations over a smaller subset of the lake would be problematic, as temperatures from shallow and deep portions of the reservoir are averaged together.

12. Figure 5: Why was Diff_3 included here and no other sensitivity run?

13. P. 15, line 10: "by as much as $\sim$ 1.3 C" ?

14. Table 4: Coloring indicates the smallest and largest absolute values.

15. P. 17, line 9: "in top 10-m temperatures"

16. P. 18: Consider including RMSE or other error metric here, as done in the previous section, as Diff_1 and 2 both contain over and underestimates of temperatures in the profiles and a quantitative measure would be valuable to the reader.

17. Figures 8 & 9, 10 & 11: Keep coloring for runs consistent between plots.

18. Figure 9: The logarithmic axes here makes it hard to put the simulated values in context with the observations from Li (1973). Consider using gray shading in the background to plot the observed range directly on the figure for comparison.

19. P. 21: Use "are fixed to 1 mm (Rou_1)" on line 7 and " at 10 mm (Rou_2)" on line 10 for greater clarity.

20. P. 23, line 2: "minimal changes to LSTs"

References:

Gula, J. and Peltier, W. R.: Dynamical downscaling over the Great Lakes basin of North America using the WRF regional climate model: the impact of the Great Lakes system on regional greenhouse warming, J. Climate, 25, 7723–7742, 2012.

Mallard, M. S., Nolte, C. G., Bullock, O. R., Spero, T. L., and Gula, J.: Using a coupled lake model with WRF for dynamical downscaling, J. Geophys. Res., 119, 7193–7208,

doi:10.1002/2014JD021785, 201

---

## Author Comment (AC1) · 17 Dec 2018

Please refer to the zip file in the Supplement for our response. The files consist of: 1. a point-to-point response file; 2. an annotated manuscript with the updated sentences/figures displayed in BLUE font; 3. a clean manuscript

Please also note the supplement to this comment:
https://www.geosci-model-dev-discuss.net/gmd-2018-168/gmd-2018-168-AC1-supplement.zip

---

## Author Response (AR1)

**Response to Reviewer 1**
* * *
*This manuscript presents a study striving to improve and calibrate an existing 1D lake model (parameterization) applied to a large artificial reservoir in China. Though the paper falls well into the journal scope, the main problems of the paper (lack of scientific novelty and methodological drawbacks) call for major revisiting of the whole study.*

**Response:**

We thank the reviewer for all his/her efforts in evaluating our work. We have prepared point-to-point responses and revised the manuscript carefully with detailed changes (in blue) given below.

*1. The base version of the model used is WRF-Lake, used in a number of studies during recent years (Gu et al., 2015; Gu et al., 2016; Xiao et al., 2016). The authors introduce into the model physical parameterizations already tested in other 1D models, especially CLM4-LISSS (Subin et al., 2012). Increasing the vertical resolution of the model is rather a technical improvement, as it is quite evident a-priori, that having 10 numerical layers is a rough resolution for deep lakes, where only 2-3 layers would cover the mixed layer. The authors follow the same approach as in (Gu et al., 2015; Gu et al., 2016; Xiao et al., 2016) to tackle additional mixing in thermocline, which is just to multiply the molecular diffusivity by 100 or other large calibrated multipliers. This is the simplest way which does not take into account the evident physical effects like suppression of mixing by stratification. The approach by Fang and Stefan (1996) to parameterize background diffusivity takes into account stratification (eq. (9)), but the authors reject it. They argue, that original eq. (9) provides insufficient mixing. I would expect, that true development of the model physics would mean to replace primitive calibration of constant multiplier by calibration of constants in eq. (9) or likewise still simple but physically-sound parameterizations.*

**Response:**

First of all, we would like to thank the reviewer for his/her insightful and constructive comments and critique, which provide us an opportunity to elaborate our thinking. We are really grateful for that. The reviewer's comments can be essentially summarized as follows:

1) *The reviewer considers the modification of vertical resolution a trivial technique to improve the model performance.*

2) *The reviewer deems the approach of this study to adjust thermal diffusivity unphysical (e.g., neglect of suppression of mixing by stratification) and asked the authors to propose a physically-sound parameterization following the formulation by Fang and Stefan (1996) (which the reviewer thought the authors rejected in this study).*

Our point-to-point responses are as follows:

1) The refinement is NOT trivial in both the technical implementation and model performance. For the technical implementation, we did NOT simply increase the vertical resolution; instead, we adopted a mechanism based on fixed factor (FF) to allow layer thicknesses adaptively increase with depth, which guarantees a smooth thickness change and thus better numerical stability. For the model performance, we clearly demonstrated the outperformance of this new discretization scheme in

simulating temperature profiles compared with the original scheme (cf. Figure 7). Also, we note this is for the first time an adaptive discretization scheme being introduced to WRF-Lake. Also, similar adaptive discretization techniques are widely used by various geoscientific models to improve the model performance (e.g., grid stretching in GFDL HiRAM (Harris et al., 2016), nonuniform meshing in MPAS (Skamarock et al., 218), etc.), which can hardly be considered as trivial modifications in model development.

2) First, we CANNOT agree with the reviewer that our approach is unphysical because this approach is based on Gu et al. (2015), whose parameterization of $k_e$ DOES consider the suppression of mixing by stratification and depth (cf. equation A1). Second, we did NOT reject the Fang and Stefan (1996) approach but actually adopted their formulation of the enhanced term $D_{ed}$ (i.e., equation (9): $D_{ed} = 1.04 \times 10^{-8}(N^2)^{-0.43}$) to account for the unresolved 3D diffusion. However, as Subin et al. (2012) pointed out, $D_{ed}$ is of the same order of magnitude of molecular diffusivity and may not make up for underestimated diffusivity. Also, a thorough calibration of empirical constant in $D_{ed}$ (i.e., $1.04 \times 10^{-8}$) is infeasible for the deep reservoir examined in this study as the necessary water temperature observations of very fine temporal and spatial resolutions were unavailable during the study period. As such, we adopted a compromised but effective approach by enlarging the constant with a factor of 100, which produced overall diffusivity similar to that measured at Lake Zürich (Li, 1973), a lake of similar topography and depth: see Figure R1 (a reprint of Figure 9 in the manuscript) and Figure R2.

[Figure]

**Figure R1.** Monthly vertical diffusivity profile for the first 60 m water by BL (black line), Diff_1 (red line), Diff_2 (blue line), and Diff_3 (gray line) in year 2015. The gray shading

indicates the diffusivity range of Lake Zürich reported by Li (1973).

[Figure]

**Figure R2.** Diffusivity profiles of different months from a) observations reported in Li (1973) and b) simulations of this study.

Besides, we deem a simple but physically-sound parameterization is necessary for the WRF-Lake model as the reviewer expected, which we aim to propose based on new observations that are being collected at our study site, the Nuozhadu Reservoir.

*2. Radiation parameterization (eq. (2)) assumes that shortwave radiation starts to decay with depth only below top 0.6 m, which is unphysical and easy to fix, assuming non-PAR (photosynthetically-active radiation) radiation to be absorbed at the surface, and PAR to be attenuated immediately below (and it is done in a such a way in almost all 1D lake models).*

**Response:**

We thank the reviewer's suggestion for a PAR-based separation scheme, and we deem a more site-specific cutoff depth for WRF-Lake (which is currently set as 0.6 m) is arguably needed. However, we CANNOT agree with the reviewer that the design for a cutoff depth by the current radiation parameterization in WRF-Lake (Bonan, 1995; Subin et al., 2012) is unphysical.

We deem the essential components in a physical radiation parameterization for water body should at least include:

1) An intensity-decaying formulation as a function of penetration depth following the Beer–Lambert law (Jerlov, 1976).
2) A wavelength-based scheme for absorption coefficients.

And we deem the current radiation parameterization in WRF-Lake does include all the two essential components: the first component is formulated with a cutoff at a certain depth (e.g., 0.6 m in the current WRF-Lake) and second adopts a simplified parameterization of absorption coefficients (cf. Deng et al. (2013) for further discussion on the depth impacts on simulated absorption).

By comparing the WRF-Lake radiation scheme with a more sophisticated 9-band scheme (Paulson and Simpson, 1981) (Figure R3), we can see that 1) the cutoff depth set by WRF-Lake essentially incorporates absorption effects of all minor bands (i.e., band 3 – 9) in the topmost part and 2) the simplified parameterization of absorption coefficients demonstrates

good consistency with Paulson and Simpson (1981) scheme in the absorption effects of major bands (i.e., band 1 and 2) at deeper depths.

As such, we deem the current radiation parameterization in WRF-Lake is physical and performs reasonably well in capturing the absorption effects of water bodies.

[Figure]

**Figure R3.** Shortwave radiation absorption simulated by WRF-Lake and Paulson and Simpson (1981). *WRF-Lake* applies the same extinction coefficient as the manuscript (i.e., $\eta = 1.00 \ m^{-1}$). *WRF-Lake (band 1)* and *WRF-Lake (band 2)* applies the same extinction coefficient as band 1 and band 2, representatively.

*3. I also see a notable drawback of the paper in that no empirical constrains have been involved on water turbidity for this particular lake. I can hardly imagine that no Secchi disk measurements have been performed at all. Treating both radiation extinction coefficient and background diffusivity which are the main controls for vertical distribution of heat as calibration parameters, you may attain similar vertical temperature distributions at different combinations of those parameters, and what would be the physical sense of that?*

**Response:**

We thank the reviewer for this advice, which would be very helpful if any improvement in radiation scheme of WRF-Lake should be implemented. However, given the infeasibility of deployment of Secchi disk measurements in our study site (a very deep reservoir under frequent operation), the improvement of radiation scheme in WRF-Lake had to be on hold in this study. We also note a similar difficulty undergone by Gu et al. (2015) who ended up with default parameterization for light extinction coefficient as well.

*4. The reservoir exhibits drastic surface level changes (about 30 m!), which would certainly influence the temperature profiles and introduce the vertical velocity in eq. (4), but the latter was not done, and the possible effects of level changes were not even discussed.*

**Response:**

We fully agree with the reviewer that the impacts of water level change on water thermal

regimes should be accounted for in WRF-Lake and we deem such impact is one of the signatures of operational reservoirs that differ from natural lakes.

As such, we have been developing a new module for the next release of WRF-rLake (not shown in this study) to take the effects of inflow and outflow into consideration (e.g., water level change). The preliminary results (Figure R4) of the next release of WRF-rLake (BL, black lines) can better reproduce the temperature profile below 20 m compared with the current release described in this study (CTL, red lines), in particular for warm months (April, May, June, and July).

[Figure]

**Figure R4.** Monthly vertical temperature profile for the first 90 m water in year 2015 by observation (red dots), BL (the next release of WRF-rLake) and CTL (WRF-rLake).

As the complete results of next release of WRF-rLake are not ready for presentation in this paper, to address the reviewer's concern, we have added discussion on the effects of inflow and outflow in the revised manuscript:

Besides, the operation-induced in/outflow, a key difference from natural lakes, is yet to be considered; given reservoirs as essential infrastructures for utilisation and management of water resources (ref), the WRF-rLake framework can be extended with operation-aware features (e.g., in/outflow parameterization) for reservoirs to better characterise the reservoir-atmosphere interactions under more realistic anthropogenic influences.

**Response:**

Thanks for the valuable suggestion. We have added more metrics as suggested in Table R1 in the revised manuscript:

BL yields the smallest RMSE of 1.13 °C against monthly observed lake temperatures profiles, while Diff_1, Diff_2, and Diff_3 yield 1.62 °C, 1.47 °C, 1.47 °C, respectively.

**Table R1.** Statistics of the discrepancy between simulated (BL, Diff_1, Diff_2, and Diff_3) and observed monthly temperature profiles during year 2015. Coral and green coloring indicate the largest and smallest absolute values among three simulations, respectively.

| | BL | Diff_1 | Diff_2 | Diff_3 |
|---|---|---|---|---|

| | | | | | |
|---|---|---|---|---|---|
| | RMSE (°C) | 1.13 | 1.62 | 1.47 | 1.47 |
| Monthly | MBE (°C) | 0.57 | -0.27 | 0.32 | 0.32 |
| Temperature | Max Bias (°C) | 3.39 | 4.56 | 5.64 | 5.63 |
| Profile | Min Bias (°C) | -1.37 | -4.32 | -3.61 | -3.58 |
| | MAE (°C) | 0.84 | 1.23 | 1.11 | 1.10 |

*17. Figures 8 & 9, 10 & 11: Keep coloring for runs consistent between plots.*

**Response:**

Corrected as suggested.

*18. Figure 9: The logarithmic axes here make it hard to put the simulated values in context with the observations from Li (1973). Consider using gray shading in the background to plot the observed range directly on the figure for comparison.*

**Response:**

Thanks for the valuable suggestion. We have added the shading in Figure 9 of the revised manuscript to indicate the observed values reported in Li (1973):

[Figure]

**Figure R1.** Monthly vertical diffusivity profile for the first 60 m water by BL (black line), Diff_1 (red line), Diff_2 (blue line), and Diff_3 (gray line) in year 2015. The gray shading indicates the diffusivity range of Lake Zürich estimated by Li (1973). (This is a reprint of Figure 9 in the manuscript)

Also, we put the simulated diffusivity of all months in one figure to compare with the data from Li (1973) for clearer illustration (Figure R2). As shown in the following figure. The left

part is observation from Li (1973), the right part is simulated by updated WRF-rLake. For most months, the observed diffusivity for the main water body ranges between 0.1~1.0 cm$^2$ s$^{-1}$ and the simulation agrees with observation well.

[Figure]

**Figure R2.** Diffusivity profiles of different months from a) observations reported in Li (1973) and b) simulations of this study.

*19. P. 21: Use "are fixed to 1 mm (Rou_1)" on line 7 and "at 10 mm (Rou_2)" on line 10 for greater clarity.*

**Response:**

Corrected as suggested.

*20. P. 23, line 2: "minimal changes to LSTs"*

**Response:**

Corrected as suggested.

[revised manuscript text omitted]

---

## Author Response (AR2)

**Response to Reviewer 1**
* * *
*This manuscript presents a study striving to improve and calibrate an existing 1D lake model (parameterization) applied to a large artificial reservoir in China. Though the paper falls well into the journal scope, the main problems of the paper (lack of scientific novelty and methodological drawbacks) call for major revisiting of the whole study.*

**Response:**

We thank the reviewer for all his/her efforts in evaluating our work. We have prepared point-to-point responses and revised the manuscript carefully with detailed changes (in blue in the main text and also) below.

*1. The base version of the model used is WRF-Lake, used in a number of studies during recent years (Gu et al., 2015; Gu et al., 2016; Xiao et al., 2016). The authors introduce into the model physical parameterizations already tested in other 1D models, especially CLM4-LISSS (Subin et al., 2012). Increasing the vertical resolution of the model is rather a technical improvement, as it is quite evident a-priori, that having 10 numerical layers is a rough resolution for deep lakes, where only 2-3 layers would cover the mixed layer. The authors follow the same approach as in (Gu et al., 2015; Gu et al., 2016; Xiao et al., 2016) to tackle additional mixing in thermocline, which is just to multiply the molecular diffusivity by 100 or other large calibrated multipliers. This is the simplest way which does not take into account the evident physical effects like suppression of mixing by stratification. The approach by Fang and Stefan (1996) to parameterize background diffusivity takes into account stratification (eq. (9)), but the authors reject it. They argue, that original eq. (9) provides insufficient mixing. I would expect, that true development of the model physics would mean to replace primitive calibration of constant multiplier by calibration of constants in eq. (9) or likewise still simple but physically-sound parameterizations.*

**Response:**

First of all, we would like to thank the reviewer for his/her insightful and constructive comments and critique, which provide us an opportunity to improve communication of our results. The reviewer's comments can be summarized into two main points:

1) *The reviewer considers the modification of vertical resolution a trivial technique to improve the model performance.*

2) *The reviewer deems the approach of adjusting thermal diffusivity to be unphysical (e.g., by neglecting the suppression of mixing by stratification) and asked the authors to propose a physically-sound parameterization following the formulation by Fang and Stefan (1996) (which the reviewer thought the authors rejected in this study).*

Our point-to-point responses are as follows:

1) Our vertical resolution refinement is not trivial in both the technical implementation and model performance. For the technical implementation, we did not simply increase the vertical resolution; instead, we adopted a mechanism (i.e., the often-observed exponentially varying structure of temperature profiles in deep water bodies) to allow layer thicknesses to adaptively increase with depth, which guarantees a smooth layer thickness change and thus better numerical stability. This approach is described in

Page 8 (Lines 1-6) of the revised manuscript. We demonstrated the improved performance of this new discretization scheme in simulating temperature profiles compared with the original scheme (cf. Figure 7). Also, we note this is the first time an adaptive discretization scheme has been applied in WRF-Lake. Similar adaptive discretization techniques are widely used by various geoscientific models to improve model performance, e.g., grid stretching in GFDL HiRAM (Harris et al., 2016) and nonuniform meshing in MPAS (Skamarock et al., 2018) etc. These approaches are considered by their authors to be important improvements in model structure. We have added text to Page 8 (Lines 6-8) to provide this context.

2) First, for diffusivity parameterization, our approach is based on Gu et al. (2015), whose parameterization of $k_e$ does consider the suppression of mixing by stratification and depth (cf. equation A1). Second, we did not reject the Fang and Stefan (1996) approach but actually adopted their formulation of the enhanced $D_{ed}$ term (i.e., equation (9): $D_{ed} = 1.04 \times 10^{-8}(N^2)^{-0.43}$) to account for unresolved 3D diffusion. However, as Subin et al. (2012) pointed out, $D_{ed}$ is of the same order of magnitude as molecular diffusivity and therefore may not make up for the underestimated diffusivity. Unfortunately, a thorough calibration of the empirical constant in the $D_{ed}$ formulation (i.e., $1.04 \times 10^{-8}$) is infeasible for the deep reservoir examined in this study as the necessary water temperature observations at very fine temporal and spatial resolutions were unavailable during the study period. As such, we adopted a compromise, yet effective, approach by increasing the constant by a factor of 100 (the baseline simulation (BL); Figure R1; a reprint of Figure 9 in the manuscript), which produced overall diffusivity similar to that measured at Lake Zürich (Li, 1973), a lake of similar topography and depth.

[Figure]

**Figure R1.** Monthly vertical diffusivity profile for the first 60 m water by the baseline simulation (BL; black line), Diff_1 (red line), Diff_2 (blue line), and Diff_3 (gray line) in year 2015. The gray shading indicates the diffusivity range of Lake Zürich reported by Li (1973).

A simple but physically-sound parameterization for the WRF-Lake model that goes beyond what we have done here is a long-term goal of our group, which, as the reviewer indicated may improve predictions. We are planning to develop such a parameterization using new observations that are currently being collected at our Nuozhadu Reservoir study site.

We thank the reviewer for these suggestions, and have added text to Page 8 (Lines 1-9) of the revised manuscript to better explain these points and clarify these important caveats.

*2. Radiation parameterization (eq. (2)) assumes that shortwave radiation starts to decay with depth only below top 0.6 m, which is unphysical and easy to fix, assuming non-PAR (photosynthetically-active radiation) radiation to be absorbed at the surface, and PAR to be attenuated immediately below (and it is done in a such a way in almost all 1D lake models).*

**Response:**

We thank the reviewer for the suggestion to (1) remove the 0.6 m assumption and (2) include a PAR-based separation scheme.

For the first point, we also questioned the reliability of the WRF-Lake default value for the base of the surface absorption layer ($z_a = 0.6$ m). Thus, in our original analyses we tested setting $z_a$ to 0, 0.1, and 0.6 m (the default value). However, these three values resulted in almost no difference in water temperature profiles at the Nuozhadu Reservoir, so we simply retained the default value of $z_a$. However, to address the reviewer's concern, we have set $z_a$ to be 0 m in WRF-rLake and reproduced all the figures with that value (there were no discernible differences in any of the figures).

For the second point, we note that the current radiation parameterization in WRF-Lake includes (a) an intensity-decaying formulation as a function of penetration depth following the Beer–Lambert law (Jerlov, 1976), with a revised 0 m cutoff depth and (b) a wavelength-based scheme for absorption coefficients based on a simplified parameterization of absorption coefficients (cf. Deng et al. (2013) for further discussion on the depth impacts on simulated absorption). Although separating radiation into non-PAR and PAR bands with a more mechanistic approach could be valuable, we note that many 1D lake models (Minlake: Fang and Stefan, 1996; LAKE: Stepanenko and Lykosov, 2005; CLM4-LISSS: Subin et al., 2012) and even more complicated 3D lake models (Delft3D FLOW: Hydraulics, 2003) do not currently make this distinction, and that doing so requires further information about dynamic turbidity and biological activity, which can add further model uncertainty in absence of observational constraints at our studied lake (Cristofor et al., 1994). Thus, we choose to keep the current radiation scheme in this version, but have added discussion on the potential benefits of more sophisticated approaches (such as that suggested by the reviewer) in section 2.1.2 and section 4.6 (the second paragraph) of the revised manuscript.

*3. I also see a notable drawback of the paper in that no empirical constrains have been involved on water turbidity for this particular lake. I can hardly imagine that no Secchi disk measurements have been performed at all. Treating both radiation extinction coefficient and background diffusivity which are the main controls for vertical distribution of heat as calibration parameters, you may attain similar vertical temperature distributions at different combinations of those parameters, and what would be the physical sense of that?*

**Response:**

We thank the reviewer for this advice, which would be very helpful if any improvement in the WRF-Lake radiation scheme were implemented. Unfortunately, no Secchi disk measurements were available for our study site and period. We note similar issues in the recent work by Gu et al. (2015), who also applied the default parameterization for light extinction coefficient.

We updated Page 15 Lines 8-14 in the revised manuscript to explain our choice of light extinction coefficient:

Although the default parameterization of light extinction coefficient has been applied in previous WRF-Lake studies (e.g., Gu et al., 2015), we tested the impacts of different values of this coefficient. Given that no Secchi disk measurements were available in our study site, no empirical constrains for the light extinction coefficient could be directly developed. Thus, we tested a range of light extinction coefficient values: 0.13 $m^{-1}$ (default), 0.30 $m^{-1}$, 1.00 $m^{-1}$, and 3.00 $m^{-1}$. Although measurements have reported larger variability of light extinction coefficient (e.g., 0.05 to 7.1 $m^{-1}$ in Subin et al. (2012)), we found simulated temperature profiles were insensitive to values outside of the 0.13 to 3.0 $m^{-1}$ range.

*4. The reservoir exhibits drastic surface level changes (about 30 m!), which would certainly influence the temperature profiles and introduce the vertical velocity in eq. (4), but the latter was not done, and the possible effects of level changes were not even discussed.*

**Response:**

We agree with the reviewer that the impacts of water level change on water thermal regimes should be accounted for in WRF-Lake and we recognize that these dynamics are important features of operational reservoirs that differ from natural lakes. As such, our future work includes developing a new module for the next release of WRF-rLake (not shown in this study) to take the effects of inflow and outflow into consideration (e.g., water level change). However, to address the reviewer's concern, we have added a new section (4.6: Uncertainties and limitations) to discuss the effects of inflow and outflow:

Operation-induced inflows and outflows are key features of artificial reservoirs and can strongly affect seasonal and interannual evolution of reservoir surface water levels, intensity of thermal stratification, and thermal structure (Anohin et al., 2006; Çalışkan and Şebnem, 2009). Given that reservoirs are essential infrastructures for utilisation and management of water resources (Jain and Singh, 2003; Ahmad et al, 2014), the WRF-rLake framework should be extended to include reservoir operation features (e.g., inflow and outflow controls) to better characterize reservoir-atmosphere interactions.

**Reference:**

Çalışkan, Anıl, and Şebnem Elçi.: Effects of selective withdrawal on hydrodynamics of a stratified reservoir. Water Resources Management, 23(7), 1257-1273, 2009.

Deng, B., Liu, S., Xiao, W., Wang, W., Jin, J., & Lee, X.: Evaluation of the CLM4 lake model at a large and shallow freshwater lake*, Journal of Hydrometeorology, 14(2), 636-649, 2012.

Fang, X., & Stefan, H. G.: Long-term lake water temperature and ice cover simulations/measurements, Cold Regions Science & Technology, 24(3), 289-304, 1996.

Harris, L. M., Lin, S.-J. and Tu, C.: High-Resolution Climate Simulations Using GFDL HiRAM with a Stretched Global Grid, Journal of Climate, 29(11), 4293–4314, doi:10.1175/jcli-d-15-0389.1, 2016.

Hydraulics. (2003). Delft3D-FLOW: simulation of multi-dimensional hydrodynamic flows and transport phenomena, including sediments—user manual.

Skamarock, W. C., Duda, M. G., Ha, S. and Park, S.-H.: Limited-Area Atmospheric Modeling Using an Unstructured Mesh, Monthly Weather Review, 146(10), 3445–3460, doi:10.1175/mwr-d-18-0155.1, 2018.

Jerlov N. G.: Marine optics. Elsevier Scientific Publishing Company, Amsterdam, The Netherlands, 231 pp, 1976.

Paulson, C. A. and Simpson, J. J.: The temperature difference across the cool skin of the ocean, Journal of Geophysical Research, 86(C11), 11044, doi:10.1029/jc086ic11p11044, 1981.

*Overall comments:*

*The manuscript describes several modifications to the lake module within WRF, which are systematically included in a series of experiments, ultimately showing that these modifications result in improved model performance within a large, deep reservoir. Overall, these modifications are explained and justified well, and it is encouraging to see that they result in more accurate simulation of surface temperatures and in more realistic temperature profiles. I have identified mostly minor issues which are outlined below, and the paper should be accepted once these are properly addressed.*

**Response:**

We appreciate the reviewer's appreciation of this work and sincerely thank him/her for these in-depth comments. We have prepared point-to-point responses and revised the manuscript carefully with detailed changes (in blue) given below.

*1. The one notable drawback to this paper is that there is no evaluation of simulated ice coverage, which can be a significant factor in how some lakes interact with the atmosphere. Although this follows naturally from the fact the that the reservoir being evaluated does not appear to experience freezing temperatures, this may limit the applicability of these results to large, deep lakes that do experience freezing, such as the Great Lakes. This limitation should be discussed.*

**Response:**

We agree with the reviewer that the ability in simulating ice coverage dynamics, or lack thereof, should be a key feature of lake models. However, as noted by the reviewer, the Nuozhadu Reservoir does not experience any ice-covered periods. Therefore, in this study, WRF-rLake could not be tested for its ability to simulate ice cover dynamics. We have added text to the revised manuscript (section 4.6: first paragraph) to address these issues:

Ice and snow processes could play a significant role in lake-atmosphere interactions (Brown and Duguay, 2010), especially for high-latitude lakes (e.g., North Eurasian lakes (Subin et al., 2012), Great Lakes (Xiao et al., 2016)). However, given the warm climatology of the Nuozhadu Reservoir, we only examined here the performance of WRF-rLake under ice-free conditions. Future work should be carried out to assess WRF-rLake performance at more reservoirs or lakes with ice-covered periods as well as different bathymetry and climate to evaluate the broader model applicability.

*2. It should be clarified early on that this evaluation of the lake model in WRF is done with observed forcing data, instead of model simulated fields. I understand that the authors' intent is most likely to evaluate the lake module free from bias that may be present in the WRF-simulated fields. However, as the model is referred to as WRF- Lake, readers may assume that the coupled system is being evaluated here. The need for such analysis isn't even mentioned until the last line of the paper, but would be better placed much earlier on.*

**Response:**

Thanks for the advice and this point is now clarified in section 3.2 of the revised manuscript as follows:

We ran the lake module off-line, driven directly by forcing data acquired from local meteorological stations rather than WRF-simulated fields, in order to evaluate the lake module free from potential biases originating in WRF.

*3. Several figures (1, 2, 3) are never referenced in the text.*

**Response:**

They are now referenced in the revised manuscript.

*4. Here, observed water temperature profiles are used and the description of the experiments implies that no spin-up time was given to the model. This differs from the practice of many other modeling efforts where observed profiles of lake temperatures are not available, some of which use larger domains that include multiple lakes. In such applications, a sufficiently long spin-up would be the only way to obtain realistic temperature profiles. Clarify whether spin-up was used and discuss the implication for your results.*

**Response:**

We did conduct a spin-up for seven days before the analysis period. This is now clarified in the revised manuscript (Page 12 Line 2).

*specific comments*

*1. P. 2, line 11: During this time of the year, snow is enhanced around the Great Lakes, not reduced.*

**Response:**

We have clarified in the revised manuscript that this conclusion (i.e., snow is reduced during fall and early winter) by Long et al. (2007) is only applicable to northern lakes or high-latitude lakes like the Great Bear Lake, rather than the Great Lakes (Page 2 Line 11):

During fall and early winter, when the lake surface is warmer than the overlying air, high-latitude lakes (e.g., the Great Bear Lake in Canada) release the heat collected during summer to the atmosphere, reducing snow accumulation in the surface areas around the lakes (Long et al., 2007).

*2. P. 3: Works by Gula et al. (2012) and Mallard et al. (2014) (which coupled WRF to FLake in 1-way and 2-way model configurations, respectively) should be briefly mentioned alongside the discussion of the FLake model in the introduction, as it is the only other lake model that has been coupled with WRF.*

**Response:**

Thanks for providing us with the importance references, which have been added in the revised manuscript (Page 3 Line 7-8).

*3. P. 7, line 7, "approximately 10%" as 90% is included plus the 0.1 m first layer.*

**Response:**

Corrected as suggested.

*4. P. 8, first paragraph: Relationship between SH and LH and Zom is not well-explained in the earlier referred to section. Subin et al. (2012) contains equations that do relate the fluxes to aerodynamic resistance, and I suggest pointing readers to the appropriate section so they can find a more thorough discussion.*

**Response:**

Rephrased as suggested:

Section 2.1.1: A more thorough discussion of the relationship between lake surface fluxes and aerodynamic resistances is provided by section 2.1.8 in Subin et al. (2012).

Section 2.2.2: As discussed in section 2.1.1, the aerodynamic resistances for heat ($r_{ah}$) and vapor ($r_{aw}$) heat fluxes are critical for surface energy balance predictions. The aerodynamic resistances are functions of momentum ($z_{0m}$) and scalar roughness lengths ($z_{0h}$ for sensible heat and $z_{0q}$ for latent heat).

*5. P. 8, line 18: This modification for frozen lakes does not appear well-justified.*

**Response:**

We thank the reviewer for this comment. The modification for frozen lakes is also adopted from Subin et al. (2012), which is similar to values reported by Andreas (1987), Morris (1989) and Vavrus et al. (1996). We agree that the parameterization of roughness lengths for frozen lakes should also be improved and justified. However, the Nuozhadu Reservoir is unfrozen throughout the year, so we did not make any adjustments for the effects of lake ice. We expect in future work the revised model would be applied to other reservoirs with frozen periods, which will allow more tests to be carried out to further justify the parameterization of roughness lengths.

The following sentence is added to the last paragraph of section 2.2.2:

It is worth noting that the parameterization of roughness lengths for frozen lakes could also be improved. However, as the Nuozhadu Reservoir is unfrozen throughout the year, we did not make any modifications to the representations for lake ice. Future work should investigate lakes with frozen periods to further improve the roughness length parameterization.

*6. P. 9, last paragraph: K is stated to be lake dependent, but a constant for it is then specified. Does K need to be provided in each lake or is it assumed to be equal to the provided constant? Also, clarify whether the Kx100 modification is applied everywhere in lakes deeper than 50 m or if it is only applied below 50 m.*

**Response:**

K is empirical and prescribed rather than lake dependent (Fang and Stefan, 1996). We are sorry about the confusion brought up by our previous statement. We have replaced K by directly applying the constant $1.04 \times 10^{-8}$.

We also clarified the statement in section 2.2.3:

Therefore, for lakes deeper than 50 m, we imposed an increase in $D_{ed}$ by a factor of 100 for all layers and argue that more analyses are required to robustly represent unresolved turbulence.

*7. P. 10. It is stated that this reservoir provides a good example of the impacts of artificial water bodies on regional climate, but this focus is not put in further context. Why did the authors choose to study an artificial body instead of a natural one?*

**Response:**

We thank the reviewer for bringing up the very valuable concern on the broader impacts of this study (i.e., influence on regional climate of human exploitations of water resources). We have added discussion to the revised manuscript regarding the uniqueness of artificial reservoirs compared to natural lakes (section 4.6: Uncertainties and limitations). Please refer to our response to the 4th comment by Reviewer 1.

We also noted in the revised conclusion that:

Our future work will couple the WRF-rLake module with the WRF framework to examine the performance of the coupled system.

*8. P. 11 first paragraph: As the LW and SW data are interpolated from 3 hourly observations, peak radiation values may be underestimated. This should be stated in the text.*

**Response:**

Discussion on the underestimated peak values are now added to Page 12 Line 5-6:

Although it probably underestimates peak radiation values, linear interpolation may still be considered to be an acceptable approximation given no data of higher temporal resolution is available.

*9. Fig. 4. Label the y-axis. Also, clarify that the "water level" shown (according to the inset box) is not actually the water level (which, having a mean of 812 m, does not seem to be consistent with the values shown here).*

**Response:**

The y-axis issue has been fixed in the revised Figure 4.

For the water level, "812 m" mentioned in P. 10, line 14 refers to the "normal water level" of the reservoir, rather than average water level. For a reservoir whose outflow is controlled wholly or partially by movable gates, normal water level is the maximum level to which water may rise under normal operation conditions. So, it is normal for the water level to fall below 812 m throughout the year of 2015. The explanation of the term "normal water level" is added as a footnote in section 3.1.

*10. Table 3 "Roughness Lengths" column: I believe the constants given here refer to the roughness lengths for unfrozen lakes, based on previous discussion, but this should be clarified.*

**Response:**

Clarified as suggested in a new note for Table 3.

*11. Figure 5 and other similar figs: The observed temperatures shown here were taken near the dam of the reservoir. Are the simulated LSTs taken and averaged over a similar area or are they representative of lake-average conditions? If it's the latter, then direct comparison to observations over a smaller subset of the lake would be problematic, as temperatures from shallow and deep portions of the reservoir are averaged together.*

**Response:**

The former: we used simulation results of an area near the dam, where the observations were collected, to conduct the evaluation. This point is now clarified in the revised manuscript as follows:

Section 4.1: The simulation results near the dam, the same place where the observations were collected, were used to conduct the evaluation.

*12. Figure 5: Why was Diff_3 included here and no other sensitivity run?*

**Response:**

This was intentionally chosen for better legibility: Diff_1, Diff_2 and Diff_3 would overlap if put together as their differences are more pronounced in temperature profiles compared with the surface temperature (cf. Figure 8). This is now clarified in the revised manuscript:

Section 4.1: Here the results of other diffusivity experiments (i.e., Diff_1 and Diff_2) are not shown.

*13. P. 15, line 10: "by as much as ∼1.3C"?*

**Response:**

The phrase "up to" is added to make the statement more precise.

*14. Table 4: Coloring indicates the smallest and largest absolute values.*

**Response:**

Colored as suggested.

*15. P. 17, line 9: "in top 10-m temperatures".*

**Response:**

Corrected as suggested.

*16. P. 18: Consider including RMSE or other error metric here, as done in the previous section, as Diff_1 and 2 both contain over and underestimates of temperatures in the profiles and a quantitative measure would be valuable to the reader.*

**Response:**

Thanks for the valuable suggestion. We have added more metrics as suggested in Table R1 in the revised manuscript:

BL yields the smallest RMSE of 1.13 °C against monthly observed lake temperatures profiles, while Diff_1, Diff_2, and Diff_3 yield 1.62 °C, 1.47 °C, 1.47 °C, respectively.

**Table R1.** Statistics of the discrepancy between simulated (BL, Diff_1, Diff_2, and Diff_3) and observed monthly temperature profiles during year 2015. Coral and green coloring indicate the largest and smallest absolute values among three simulations, respectively.

|  |  | BL | Diff_1 | Diff_2 | Diff_3 |
|---|---|---|---|---|---|
|  | RMSE (°C) | 1.13 | 1.62 | 1.47 | 1.47 |
| Monthly | MBE (°C) | 0.57 | -0.27 | 0.32 | 0.32 |
| Temperature | Max Bias (°C) | 3.39 | 4.56 | 5.64 | 5.63 |
| Profile | Min Bias (°C) | -1.37 | -4.32 | -3.61 | -3.58 |
|  | MAE (°C) | 0.84 | 1.23 | 1.11 | 1.10 |

*17. Figures 8 & 9, 10 & 11: Keep coloring for runs consistent between plots.*

**Response:**

The coloring is now made consistent between these plots.

*18. Figure 9: The logarithmic axes here make it hard to put the simulated values in context with the observations from Li (1973). Consider using gray shading in the background to plot the observed range directly on the figure for comparison.*

**Response:**

Thanks for the valuable suggestion. We have added the shading in Figure 9 of the revised manuscript to indicate the observed values reported in Li (1973). Also, please refer to our response to the first comment by Reviewer 1 (Figure R1).

*19. P. 21: Use "are fixed to 1 mm (Rou_1)" on line 7 and "at 10 mm (Rou_2)" on line 10 for greater clarity.*

**Response:**

Corrected as suggested.

*20. P. 23, line 2: "minimal changes to LSTs"*

**Response:**

Corrected as suggested.

**Reference:**

Andreas, E. L.: A theory for the scalar roughness and the scalar transfer coefficients over snow and sea ice, Boundary-Layer Meteorology, 38(1-2), 159–184, doi:10.1007/bf00121562, 1987.

Morris, E.: Turbulent transfer over snow and ice, Journal of Hydrology, 105(3-4), 205–223, doi:10.1016/0022-1694(89)90105-4, 1989.

Vavrus, S. J., Wynne, R. H. and Foley, J. A.: Measuring the sensitivity of southern Wisconsin lake ice to climate variations and lake depth using a numerical model, Limnology and Oceanography, 41(5), 822–831, doi:10.4319/lo.1996.41.5.0822, 1996.

**Response to Editor**

Dear Editor:

We thank you for the opportunity to respond to the reviewer concerns. In previous parts of this document, we have prepared detailed responses to all of the comments, and feel that doing so has substantially improved communication of our results in the revised manuscript.

*Reviewer #1:*

*Reviewer #1 has indicated a number of concerns which do not seem to have been addressed in the revision. I recommend you further review these comments. Note that further peer review may then be required.*

**Response:**

Please see our detailed responses to each of Reviewer #1's concerns in this document.

*1) No changes seem to have been made to the paper to acknowledge this comment noting, for example, that you change both the number of layers and their distribution, and that there are many ways in which this could be done.*

**Response:**

Please see our response to Reviewer #1's first comment. Changes are now made to section 2.2.1 in the revised manuscript to better explain and justify our approach. The updated text is annotated and related references have been added to the manuscript.

*2) I think the text around equation (9) needs to be clarified. The final sentence of page 9 is particularly unclear.*

**Response:**

The units for equation (9) are specified in the text, and the last sentence was updated with a clearer statement.

*2. Discussion has been added to the response but not the paper.*

**Response:**

The following discussion is added in the revised manuscript to the last paragraph of section 2.1.2:

Though there exist more sophisticated radiation schemes in other lake models (e.g., the 9-band scheme by Paulson and Simpson, 1981), we kept the current WRF-Lake radiation scheme since it includes the essential components in a waterbody physical radiation parameterization: an intensity-decaying formulation as a function of penetration depth following the Beer–Lambert law (Jerlov, 1976) and a scheme for absorption coefficients. Such an approach is also accepted by many other 1D lake models (Fang and Stefan, 1996; Stepanenko and Lykosov, 2005). To improve the model performance, we tentatively set the cutoff depth $z_a$ to be 0 m in this version as 0.6 m is usually an overestimated value, especially for shallow lakes (Deng et al., 2013). Although adopting this $z_a$ value (0 m) demonstrates

acceptable performance in this work, a more lake-specific cutoff depth may be needed for better model performance.

We also updated our response to the reviewer regarding this point.

*3. You have acknowledged the drawback in the response but not the paper.*

**Response:**

The text of the fifth paragraph in section 3.4 of the revised manuscirpt has been updated to acknowledge this omission:

Light extinction coefficient ("Ext" set): through model tests, we conclude that in addition to the schemes we modified, the light extinction coefficient is also a key parameter for accurately modelling deep lakes (Hocking and Straskraba, 1999). Although the default parameterization of light extinction coefficient has been applied in previous WRF-Lake studies (e.g., Gu et al., 2015), we tested the impacts of different values of this coefficient. Given that no Secchi disk measurements were available in our study site, no empirical constrains for the light extinction coefficient could be directly developed. Thus, we tested a range of light extinction coefficient values: 0.13 $m^{-1}$ (default), 0.30 $m^{-1}$, 1.00 $m^{-1}$, and 3.00 $m^{-1}$. Although measurements have reported larger variability of light extinction coefficient (e.g., 0.05 to 7.1 $m^{-1}$ in Subin et al. (2012)), we found simulated temperature profiles were insensitive to values outside of the 0.13 to 3.0 $m^{-1}$ range. We concluded that the best performance could be achieved by increasing the light extinction coefficient to ~1.00 $m^{-1}$, which thus is adopted by our baseline run (BL).

*4. You have added only a very short paragraph in response to the reviewer comment.*

**Response:**

In our current response and revised manuscript, we have added more detail in a new section (4.6: Uncertainties and limitations) to discuss the effects of inflow and outflow:

Operation-induced inflows and outflows are key features of artificial reservoirs and can strongly affect seasonal and interannual evolution of reservoir surface water levels, intensity of thermal stratification, and thermal structure (Anohin et al., 2006; Çalışkan and Şebnem, 2009). Given that reservoirs are essential infrastructures for utilisation and management of water resources (Jain and Singh, 2003; Ahmad et al, 2014), the WRF-rLake framework should be extended to include reservoir operation features (e.g., inflow and outflow controls) to better characterize reservoir-atmosphere interactions.

*Reviewer #2:*

*1. The quoted text in the response is not as it appears in the paper.*

**Response:**

    Thanks for this reminder. We have added a new section 4.6: Uncertainties and limitations, to acknowledged the drawback of evaluation without ice coverage period:

    Ice and snow processes could play a significant role in lake-atmosphere interactions (Brown and Duguay, 2010), especially for high-latitude lakes (e.g., North Eurasian lakes (Subin et al., 2012), Great Lakes (Xiao et al., 2016)). However, given the warm climatology of the Nuozhadu Reservoir, we only examined here the performance of WRF-rLake under ice-free conditions. Future work should be carried out to assess WRF-rLake performance at more reservoirs or lakes with ice-covered periods as well as different bathymetry and climate to evaluate the broader model applicability.

    We have updated our response to the reviewer accordingly.

*Specific comments:*

*5. You do not seem to have discussed or addressed the reviewer comment in the paper.*

**Response:**

    To address this reviewer comment, the following statement is added to the last paragraph of section 2.2.2:

    It is worth noting that the parameterization of roughness lengths for frozen lakes could also be improved. However, as the Nuozhadu Reservoir is unfrozen throughout the year, we did not make any modifications to the representations for lake ice. Future work should investigate lakes with frozen periods to further improve the roughness length parameterization.

    The response to the reviewer is updated accordingly.

*6. I do not understand the current text around equation (9). This lacks units, and I find the final sentence of page 9 to be unclear.*

**Response:**

    We have added the unit for $D_{ed}$ ($m^2\,s^{-1}$) in the paper. As $N$ has a unit of $s^{-1}$, the unit for the empirical constant $1.04 \times 10^{-8}$ should be $m^2\,s^{-0.14}$. The last sentence on page 9 is further clarified in the revised manuscript.

*8. The quoted text in the response is not as it appears in the paper.*

**Response:**

    The quoted text in the response is updated to be consistent with that in the paper.

*13. I do not understand your response.*

**Response:**

    The updated response is as follows (response to reviewer 2):

The phrase "as much as" is added to make the statement more precise.

*17. Figures 9, 11: The caption colors do not match the legend.*

**Response:**

The captions are corrected.

*Other comments:*

*1. The article is missing an "Author Contribution" section*

**Response:**

The "Author Contribution" section is added.

---

## Author Response (AR3)

**Response to Reviewer**
* * *
*General comments:*

*In my first review I identified major deficiencies of the paper as a set of methodological drawbacks and a lack of scientific novelty. Though, I still deem the paper does not deliver a significantly new knowledge to community, I admit that the authors considerably improved the manuscript and answered my comments carefully.*

**Response:**

We appreciate the reviewer's recognition of our work in the revision of this manuscript.

*Specific comments:*

*1. I recommend to avoid naming the stretching numerical grid by the term adaptive grid, as the latter is usually applied to meshes evolving in time.*

**Response:**

Removed as suggested.

*2. I realize now that I misunderstood the setup of background diffusivity in the original version of manuscript and that the authors indeed used Fang and Stefan (1996) formulation for background diffusivity. Note, however, that expression (9) of the manuscript originally includes the lake surface area, missing in the current text. Please also consider choosing the proper reference: formula (9) was first suggested by Hondzo and Stefan (1993), not in Fang and Stefan (1996).*

**Response:**

We thank the reviewer for this reminder. We also noticed this equation should have included lake area. However, unlike the Minlake model developed by Riley and Stefan (1988), the WRF-Lake model does not take lake area into consideration. A similar issue exists in CLM4-LISSS, which adopted the values measured at an ice-covered lake (Fang and Stefan, 1996) for equation (9) by setting $\delta = 1.04 \times 10^{-8}$.

We adopted the same treatment as CLM4-LISSS but admit that this coefficient may vary from lake to lake (partially explained by lake area) and thus need to be tuned under specific scenarios. The text in the manuscript has been updated with the correct reference as suggested (see Page 10).

**Reference:**

Riley, M. J., and Stefan, H. G.: Minlake: A dynamic lake water quality simulation model, Ecological Modelling, 43(3), 155-182, 1988.

Fang, X., & Stefan, H. G.: Long-term lake water temperature and ice cover

simulations/measurements, Cold Regions Science & Technology, 24(3), 289-304, 1996.

**Response to Editor**
* * *
*1. Is it possible to archive the version of the code described in the manuscript, e.g. in a Zenodo repository?*

**Response:**

We have archived the code associated with this manuscript in a Zenodo repository, which can be found at https://doi.org/10.5281/zenodo.2624892.

*2. Is it possible to clarify the license under which the code is available?*

**Response:**

We have chosen MIT as the license for the code and has clarified it in *Code and data availability* of the manuscript.

*3. Please also ensure that data is deposited in an appropriate repository, as described in the Copernicus data policy.*

**Response:**

The complete dataset for the year 2015 cannot be made public at present in order to comply with the confidentiality agreements between Tsinghua University and China Huaneng Group Co., Ltd. However, a one-week subset of the whole dataset can be accessed at https://doi.org/10.5281/zenodo.2624892. Also, the readers may have access to the whole dataset by emailing T. Sun (ting.sun@reading.ac.uk) with a specific request.

The following statement is added to the *Code and data availability* part of the manuscript:

The WRF-rLake source code (under MIT license) with a sample dataset can be accessed at https://doi.org/10.5281/zenodo.2624892. Instructions for running the offline model can also be found in README.md via the above link. The whole dataset used in this paper is available upon request to TS (ting.sun@reading.ac.uk).

---

## Author Response (AR4)

**Response to Editor**

*Please further clarify the license under which the code is available. The cited Zenodo repository indicates "Creative Commons Attribution 4.0 International", and includes no clear indication of the use of the MIT license. The article indicates that the software is under the MIT license.*

**Response:**

We have corrected the license information of our code and data files in Zenodo repository to MIT, which is now consistent with the statement in the manuscript.